neuroscience/psychology

fMRI, encoding, decoding, semantic representation, mental simulation

**Author for correspondence:**
David Soto
e-mail: d.soto@bcbl.eu

# Decoding and encoding models reveal the role of mental simulation in the brain representation of meaning

David Soto[1,2], Usman Ayub Sheikh[1], Ning Mei[1] and Roberto Santana[3]

[1]Basque Center on Cognition, Brain and Language, Paseo Mikeletegi 69, 2nd Floor, 20009 San Sebastian, Spain
[2]Ikerbasque, Basque Foundation for Science, Bilbao, Spain
[3]Department of Computer Science and Artificial Intelligence, University of Basque Country, Leioa, Spain

DS, 0000-0003-0205-7513; UAS, 0000-0002-0259-4725

How the brain representation of conceptual knowledge varies as a function of processing goals, strategies and task-factors remains a key unresolved question in cognitive neuroscience. In the present functional magnetic resonance imaging study, participants were presented with visual words during functional magnetic resonance imaging (fMRI). During shallow processing, participants had to read the items. During deep processing, they had to mentally simulate the features associated with the words. Multivariate classification, informational connectivity and encoding models were used to reveal how the depth of processing determines the brain representation of word meaning. Decoding accuracy in putative substrates of the semantic network was enhanced when the depth processing was high, and the brain representations were more generalizable in semantic space relative to shallow processing contexts. This pattern was observed even in association areas in inferior frontal and parietal cortex. Deep information processing during mental simulation also increased the informational connectivity within key substrates of the semantic network. To further examine the properties of the words encoded in brain activity, we compared computer vision models—associated with the image referents of the words—and word embedding. Computer vision models explained more variance of the brain responses across multiple areas of the semantic network. These results indicate that the brain representation of word

meaning is highly malleable by the depth of processing imposed by the task, relies on access to visual representations and is highly distributed, including prefrontal areas previously implicated in semantic control.

# 1. Introduction

Grounded models of semantic cognition propose that knowledge about the world is re-enacted in the same modality-specific brain systems that are involved in perceptual or action processes. For instance, the concept of 'guitar' comprises the way it looks, how it is played, and the sound it makes. Damasio's convergence zone theory proposes that re-enactment is mediated by brain association areas that integrate information from modality-specific systems [1]. The perceptual symbols theory [2] proposes that co-activation patterns in sensory-motor substrates are critical. On this account, conceptual knowledge involves an agent's brain simulating the different properties of the object in question (e.g. shape, colour, texture, sound, action) in a way that resembles how the information is encoded in sensorimotor systems during overt behaviour.

Different neurocognitive models of semantic knowledge indicate the role of 'hub' regions which are implicated in the integration of modality-specific (sensory) information and the formation of invariant conceptual representations [3]. There is, however, debate on whether a single hub [4] or multiple hubs exist [1]. For instance, the inferior parietal [5] and lateral and ventral temporal regions [6] have been also implicated in this regard. Functional magnetic resonance imaging (fMRI) studies using both decoding [7,8] and encoding models [9,10] further indicate the brain representation of meaning is highly distributed. Decoding studies indicate that the brain representation of semantic knowledge generalizes across words and images and is lateralized to the left hemisphere, involving the angular and intraparietal sulcus, and the posterior middle temporal gyrus [11].

Previous work has compared image-based models of word referents and text-based models at explaining fMRI responses when participants were required to think about the words presented on each trial [12]. Anderson and colleagues showed that image-based models were more similar to fMRI word-response patterns in visual object-selective regions (i.e. ventrotemporal and lateral occipital cortex) while text-based models explained activity in posterior parietal, lateral temporal and inferior-frontal regions. Additional research used representational similarity analyses to compare how behavioural models of conceptual properties of word items and models of the visual properties of the word referents explained blood-oxygen-level-dependent (BOLD) activity patterns when the participants were oriented by means of a feature verification task to process either the visual references or the conceptual properties [13]. They found evidence for distinct encoding of the different types of information: visual information in lateral occipital cortex, conceptual information was mediated by the temporal pole and parahippocampal cortex, and integration of visual and conceptual information in perirhinal cortex (see also [14], for results with visual objects). These results indicate the existence of distinct visual (i.e. modal) and non-visual (i.e. amodal) brain representations of semantic knowledge. Encoding models based on sensory-motor properties of word items (e.g. sound, colour, shape, manipulability) have also been shown to predict responses in the left lateral temporoparietal cortex [15]. Further, incorporating experiential features of the items (e.g. sensorimotor, emotional, interoceptive responses) to a text-based model significantly improved classification performance of sentences presented during fMRI [16]. However, the different studies either required participants to think about the words (e.g. [8,10,12,17] or perform a speeded semantic classification task (e.g. [15]) or merely pay attention to the items [7,9]. Hence, how task-related factors related to internal processing goals and strategies shape the brain representation of meaning remains to be determined. For instance, previous research has not determined the role of the depth of processing on the brain representation of concepts.

In this study, depth of processing was operationalized as the difference between covertly reading a word (shallow processing) and mentally simulating the properties of the concept (deep processing). Mental simulation here refers to the ability to imagine or re-enact modality-specific representations. Although this manipulation differs from the seminal experimental framework of 'levels of processing' [18], which used more targeted tasks tapping on semantic versus phonemic/orthographic properties judgements, our experimental manipulation is nevertheless consistent with varying depths of processing, in that mental simulation requires deeper access to the meaning of the word items, while the condition of shallow processing focuses mainly on phonological representations.

We used fMRI during a visual word recognition task in which participants either engaged in deep (i.e. mental simulation) versus shallow processing (i.e. covert reading), and then applied multivariate pattern analyses (MVPA) and encoding models to understand how the depth of processing affects the brain representation of meaning in a set of left-lateralized regions involved in visual word processing based on a previous meta-analysis [19]. The so-called semantic network responds to meaningful word input including sentences regardless of the particular sensory-motor content of the items and involves distributed areas in inferior parietal lobule, lateral temporal cortex, lateral prefrontal cortex, precuneus/posterior cingulate gyrus, parahippocampal gyrus, and medial prefrontal cortex. It is important to note that Binder *et al.* [19] also identified brain areas associated with semantic language processing in the right hemisphere, although semantic-related activity was higher in the left hemisphere. Because of this, we elected to focus on key semantic regions of interest (ROIs) of the left hemisphere in order to constrain the search space of our fMRI analyses. Some of these semantic regions have been implicated in mental imagery, in particular the posterior cingulate, lateral parietal, medial prefrontal (PFC) and superior PFC regions, showing imagery-related activity independent of imagery modality [20] and similar involvement during memory retrieval processes [21].

Regarding the encoding models, we tested word embedding and computer vision models fitted with the image referents of the words, in order to examine the properties of the corresponding brain representations, in particular, how semantic/syntactic versus visual properties associated with the words are encoded across the shallow and deep processing conditions. Critically, our experimental paradigm allowed us to test the role of task factors in encoding-related activity which, as noted above, prior studies did not address. We reasoned that if simulation processes for word concepts occur automatically in perceptual systems [22], then we would expect that computer vision models explain brain responses during semantic processing to a similar extent across the different depth of processing conditions. We further determined the extent to which different types of features (i.e. semantic/syntactic versus perceptual, i.e. visual) are encoded in brain activity. According to perceptual symbols theory, conceptual knowledge is supported by simulation processes in perceptual systems and this simulation is thought to occur automatically without the need of committing the information into a conscious working memory system [22].

Regarding the decoding analyses, we predicted that mental simulation ought to increase the decodability of the word category relative to when the words are merely read. We further assessed whether the depth of processing associated with mental simulation has a local or a generalized, distributed effect on the level of decoding accuracy across the regions of the semantic network. We further hypothesized that during mental simulation the multi-voxel patterns associated with a given word concept ought to be more generalizable with respect to examples of the same category; hence MVPA classifiers trained with a subset of the words should better predict out-of-sample words not used for training in the deep relative to the shallow processing condition. Finally, we reasoned that if explicit mental simulation supports integrated brain representations of meaning then the level of informational connectivity between key substrates of semantic network (i.e. the temporal correlations in the level of semantic decodability across brain areas) ought to increase relative to shallower processing conditions. The present study will assess this, while controlling for potential confounds due to potential differences in the temporal correlation among ROIs (i.e. functional connectivity) across the different task contexts.

# 2. Methods

## 2.1. Participants

Following informed consent, 27 participants (20–33 years, mean age: 24 ± 3, 10 males) took part in return of monetary compensation.

## 2.2. Experimental task and procedure

The experiment was programmed using Psychopy [23]. It comprised eight fMRI runs. Each trial began with a fixation period of 250 ms followed by a blank screen of 500 ms (figure 1) and then by the target visual word which was displayed for 1 s. The word was randomly selected from a pool of 18 living and 18 non-living words. Stimuli were centrally presented in white against black background in uppercase Arial font. The offset of the word was followed by a blank screen of 4 s. During the 4 s delay,

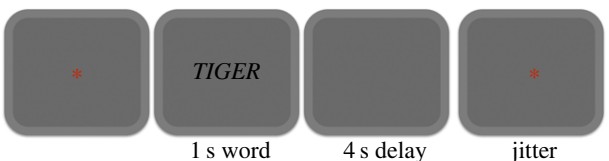

| 1 s word | 4 s delay | jitter |

**Figure 1.** Illustration of the experimental protocol.

depending on the condition, the participants were asked to either covertly read the word (shallow processing condition) or mentally simulate the properties associated with the word, henceforth, the deep processing condition. Specifically, in the shallow processing blocks, we asked participants to read and repeat the word in their mind (i.e. in the phonological loop) until the red asterisk is displayed. This was done in an attempt to equate the level of task load across task conditions. In the deep processing, participants were required to engage in mental simulation of the words, namely, they were asked to re-represent the sensory-motor experiences (i.e. item shape, colour, auditory, action- and context-related features) associated with the word during the delay period until the red asterisk is displayed. They were instructed to use similar mental simulations in subsequent presentations of the same word.

Shallow and deep processing conditions were varied across scanning runs in a counterbalanced fashion. A red asterisk was centrally presented during the inter-trial interval and participants were instructed to relax and wait for the next trial. To ensure that the participants focused on the stimuli and the task, a maximum of two catch trials were set to appear at random points in each of the sessions. These catch trials showed number words (zero, one and three) in place of usual living/non-living words, and participants were asked to respond by pressing any one of the four buttons on the fMRI response pad. The total number of catch trials was kept equal across conditions. A minimum of zero and a maximum of two catch trials were presented in each run; however, it was made sure that the total number of catch trials presented in shallow processing condition (mean = 4.05) were the same as that in the deep condition (mean = 3.85). To maximize the separation between the brain response for each of the trials, the time for which the asterisk stayed on the screen was jittered between 6 and 8 s. The jitter was based on a pseudo-exponential distribution resulting in 50% of trials with the inter-trial interval of 6 s, 25% with 6.5 s, 12.5% with 7 s and so on. The interval between word presentations across trials hence varied between 10.75 s and 12.75 s.

Each scanning run comprised 36 trials, one for each of the words, randomly selected. Living words included animal names written in Spanish and presented in uppercase, i.e. tigre, gallo, perro, oveja, cerdo, gorila, burro, yegua, ardilla, conejo, gallina, caballo, ballena, tortuga, pantera, camello, elefante, canguro. Non-living words were comprised of tool names, i.e. llave, lapiz, tijera, aguja, pinza, sierra, clavo, pincel, alicate, tuerca, navaja, cepillo, taladro, soplete, tornillo, cuchara, martillo, cuchillo. Corresponding English translations were the following. Living words: tiger, rooster, dog, sheep, pig, gorilla, donkey, mare, squirrel, rabbit, hen, horse, whale, turtle, panther, camel, elephant, kangaroo. Non-living words: wrench, pencil, scissors, needle, clamp, saw, nail, brush, pliers, nut, knife, brush, drill, blowtorch, screw, spoon, hammer, knife.

We gathered familiarity ratings for the actual Spanish stimuli itself using EsPal Database [24]. This database includes familiarity ratings between 1 and 7. There were a few words for which the rating was not found. However, if we consider only the words for which it was found, familiarity ratings were very similar (mean living: 5.94; mean non-living: 5.98). We also looked at the subjective familiarity ratings of the English translations of the same concepts. These were obtained using MRC Psycholinguistic Database [25]. This database includes ratings between 100 and 700. There were again a few words for which the rating was not found; however, we again observed that familiarity ratings are very similar between conditions (mean living: 511; mean non-living: 498).

## 2.3. MRI data acquisition

A SIEMENS Magnetom Prisma-fit scanner, with 3 T magnet and 64-channel head coil, was used to collect, for each participant, one high-resolution T1-weighted structural image and eight functional images (corresponding to eight runs/session). In each fMRI session, a multiband gradient-echo echo-planar imaging sequence with multi-band acceleration factor of 6, resolution of $2.4 \times 2.4 \times 2.4$ mm$^3$, repetition time (TR) of 850 ms, echo time (TE) of 35 ms and bandwidth of 2582 Hz Px$^{-1}$ was used to obtain 520 three-dimensional volumes of the whole brain (66 slices; field of view = 210 mm). The

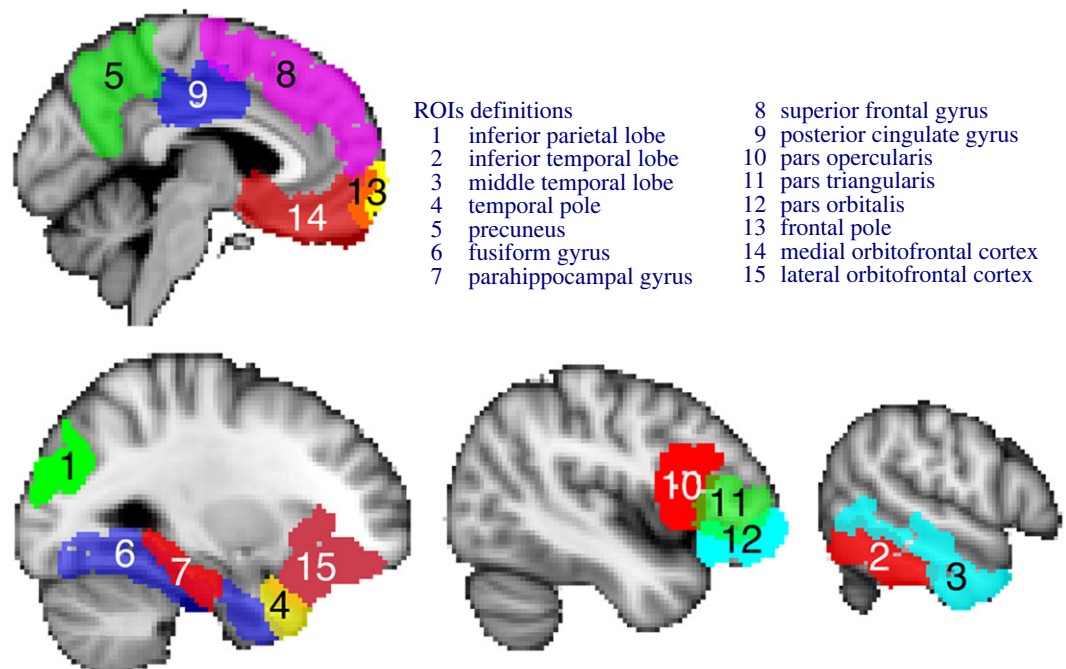

**Figure 2.** The figure shows the selected regions of interest projected on an Montreal Neurological Institute (MNI) standard template image. The 15 left-lateralized areas were pre-specified and included regions: inferior parietal lobe, inferior temporal lobe, middle temporal lobe, precuneus, fusiform gyrus, parahippocampal gyrus, superior frontal gyrus, posterior cingulate gyrus, pars opercularis, pars triangularis, pars orbitalis, frontal pole, medial orbitofrontal cortex, laterial orbitofrontal cortex and anterior temporal lobe.

visual stimuli were projected on an MRI-compatible out-of-bore screen using a projector placed in the room adjacent to the MRI-room. A small mirror, mounted on the head coil, reflected the screen for presentation to the participants. The head coil was also equipped with a microphone that enabled the participants to communicate with the experimenters in between the sessions.

## 2.4. MRI data preprocessing

The preprocessing of fMRI data was performed using FSL FEAT (FMRIB Software Library; v. 5.0). The first nine volumes were discarded to ensure steady-state magnetization; to remove non-brain tissue, brain extraction tool (BET) [26] was used; volume realignment was performed using MCFLIRT [27]; minimal spatial smoothing was performed using a gaussian kernel with full width half maximum (FWHM) of 3 mm. Next, independent component analyses-based automatic removal of motion artefacts (ICA-AROMA) was used to remove motion-induced signal variations [28] and this was followed by a high-pass filter with a cut-off of 60 s. The sessions aligned to a reference volume of the first session.

## 2.5. Regions of interest

A set of 15 left-lateralized ROIs was pre-specified (figure 2) based on a meta-analysis of the semantic system [29] and also anterior temporal lobe (ATL) due to its role in semantic cognition [3]. Hence, the ROIs included: inferior parietal lobe (IPL), inferior temporal lobe (ITL), middle temporal lobe (MTL), precuneus, fusiform gyrus (FFG), parahippocampal gyrus (PHG), superior frontal gyrus (SFG), posterior cingulate gyrus (PCG), pars opercularis (POP), pars triangularis (PTR), pars orbitalis (POR), frontal pole (FP), medial orbitofrontal cortex (MOFC), laterial orbitofrontal cortex (LOFC), and anterior temporal lobe (ATL). First, automatic segmentation of the high-resolution structural scan was done with FreeSurfer's automated algorithm `recon-all`. The resulting masks were transformed to functional space using seven d.f. linear registrations implemented in FSL FLIRT [27] and binarized. All further analyses were performed in native BOLD space.

## 2.6. Multivariate pattern analysis for decoding

Multivariate pattern analysis was conducted using scikit-learn [30] and PyMVPA [31]. Specifically, classification based on a supervised machine learning algorithm, i.e. linear support vector machine

[32] was used to evaluate whether multi-voxel patterns in each of the ROIs contained information about the word semantic category (animal versus tools) in each of the experimental contexts (i.e. deep and shallow processing).

### 2.6.1. Data preparation

For each participant, the relevant time points or scans of the preprocessed fMRI data of each run were labelled with attributes such as word, category and condition using the behavioural data files generated by Psychopy. Invariant voxels (or features) were removed. These were voxels whose value did not vary throughout the length of one session. If not removed, such features can cause numerical difficulties with procedures like $z$-scoring of features. Next, data from all sessions were stacked and each voxel's time series was run-wise $z$-scored (normalized) and linear detrended. Finally, to account for the haemodynamic lag, examples were created for each trial by averaging the six volumes between the interval of 3.4 s and 8.6 s after word onset.

### 2.6.2. Pattern classification

Linear support vector machine (SVM) classifier, with all parameters set to default values as provided by the scikit-learn package ($l$2 regularization, $C = 1.0$, tolerance = 0.0001), was used. The following procedure was repeated for each ROI separately. To obtain an unbiased generalization estimate, following [33], the data was randomly shuffled and split to create 300 sets of balanced train–test (80–20%) splits with separate items for training and testing. Each example was represented by a single feature vector with each feature being the mean of voxel intensities across the sub-interval of 3.4 s and 6.8 s. Hence, the length of feature vector was equal to the number of voxels in the ROI. To further reduce the dimensionality of the data and thus reduce the chances of overfitting [34,35], principal component analysis (PCA) with all parameters set to default values as provided by the scikit-learn was used. The number of components was equal to the number of examples thus resulting in all ROIs having an equal number of components. These components were linear combinations of the preprocessed voxel data and, since none of the components was excluded, it was an information loss-less change of the coordinate system to a subspace spanned by the examples [36]. Features thus created were used to train the decoder. Note that PCA was performed on the training set; then the trained PCA was used to extract components in the test data and its classification performance was assessed. This procedure was repeated separately for each of the 300 sets, and the mean of corresponding accuracies was collected for each of the participants.

In this first type of cross-validation, although training and test sets used independent partitions of scans, the same word example could appear in both train and test sets. Hence, we also ran a second type of cross-validation, in which the training and testing partitions used different sets of words (i.e. scans associated with a pair of words—living and non-living—were left out for testing). Hence, this second type of cross-validation involved out-of-sample generalization.

### 2.6.3. Statistics

To determine whether the observed decoding accuracy in a given ROI is statistically significantly different from the chance level of 0.5 (or 50%), $t$-tests were performed with $p$-values corrected for multiple comparisons using false discovery rate (FDR). Paired $t$-tests between decoding accuracy across ROIs in deep and shallow processing were also FDR corrected.

### 2.6.4. Informational connectivity pipeline

Informational connectivity analysis is used to identify regions of the brain that display temporal correlation in the multivariate patterns regarding the key stimulus classes [37]. The purpose of this analysis here was to investigate how this informational connectivity between the 15 pre-specified ROIs varies across the deep and shallow processing conditions. The fMRI data were preprocessed and labelled as mentioned in §§2.4 and 2.6.1. PCA was used for dimensionality reduction, and SVM for classification (see §2.6.2). A leave-one-trial-out cross-validation was performed. Specifically, the classifier was trained on all the volumes between the sub-interval of 3.4–6.8 s of all the trials except one, and was tested on all the volumes (not confined to any sub-interval) of the left-out trial. All the trials were used, one by one, as test trials starting with the first trial, and the corresponding probability of detecting a correct class was recorded for each of the volumes. In this way, a time series

of MVPA discriminability values (one per time point) was obtained for each of the ROIs. To calculate the informational connectivity between the ROIs, these time series were correlated using Pearson moment-to-moment correlation for each of the pairs of ROIs, and a matrix of informational connectivity was created. This procedure was performed separately for shallow and deep processing sessions resulting in two matrices for each of the participants.

## 2.6.5. Word embedding models

The word embedding models used in the encoding analysis (Fast Text, GloVe and Word2vec) were pre-trained [38] using the Spanish Billion Word Corpus.[1] A common feature of these three models is that they are trained based on a corpus of text, and hence are sensitive to the characteristics of the text that capture both semantic and syntactic information. For each word used in the current experiment, the corresponding vector representation of 300 dimensions (300-D) was extracted. Here 300 was a conventional choice that is commonly used in the natural language processing community. To visualize the representational patterns of the word embedding features, we computed the representational dissimilarity matrices (RDMs) of the models. Prior to the visualization, the representational feature of each word was normalized by subtracting the mean. The electronic supplementary material shows the RDMs regarding Fast Text (see figure S1), GloVe (figure S2) and Word2Vec (figure S3).

## 2.6.6. Computer vision models

We applied computer vision models to extract abstract representations of the image referents associated with the words used in the present study. VGG19 [39], MobileNetV2 [40] and DenseNet169 [41] were the computer vision models used in the current analyses. These were pre-trained models provided by the Keras Python library (Tensorflow 2.0 edition) [42] using the ImageNet dataset.[2] The pre-trained models, in general, were multi-layer convolutional neural networks. After several convolution-pooling blocks, a global pooling layer was applied to represent the image with a dense vector. Followed by several feedforward fully connected layers, the probability of each class was estimated. The training process involved predicting one of the 1000 classes of the ImageNet dataset given an image. However, the representational dimensions of the computer vision models are different from the word embedding models (e.g. VGG19 represents images with 512-D, while DenseNet represents images with 1664-D). Thus, we performed fine-tuning [43,44] to have each of the computer vision models to have 300-D as the word embedding models. The computer vision models were fine-tuned using 101 assorted object category images [45][3]. 'BACKGROUND Google', 'Faces', 'Faces easy', 'brain', 'stop sign' and 'trilobite' were removed from the categories due to the lack of conceptual relationship to the 'living, non-living' categories of our experiment. In order to balance the instances of each category, 30 images were randomly selected from each image set. Six images of each category were randomly selected to form the validation set, while the rest formed the training set. We added an additional layer of 300 units, and a classification layer to decode the rest of 96 categories. The convolution pooling blocks of the pre-trained model were frozen during training and validation, and only the newly added layer's weights were to be updated. In order to obtain robust performance, image augmentation such as rotation, shifting, zoom and flipping was applied. Before training or validating, all images were normalized according to the selected models (see preprocessing steps in Keras documentation) and resized to 128 by 128 pixels, and the normalization procedure was applied to each image individually. The activation function of the layer with 300 units was self-normalizing (SELU) [46] and the weights were initialized with LeCun normal initialization procedure [47] as suggested in the Tensorflow documentation.[4] The optimizer was Adam [48] and the learning rate was 0.0001. Loss function was categorical cross-entropy. No other regularization procedures were used. During training, the fine-tuning model could reach to a maximum of 3000 epochs with batch size of 16 images per step of back-propagation. However, if the model's performance on the validation set did not improve for five consecutive epochs, the training would be terminated and the fine-tuned weights were saved for later use.

---

[1]http://crscardellino.github.io/SBWCE/

[2]http://www.image-net.org/

[3]http://www.vision.caltech.edu/Image_Datasets/Caltech101/

[4]https://www.tensorflow.org/api_docs/python/tf/keras/activations/selu/

These fine-tuned computer vision models were applied to extract abstract representations of the image referents of the words. In particular, for each word, we sampled 10 images collected from the Internet. Images were cropped and the object appeared at the centre on a white background. The values of each image were normalized according to the selected model. The output vector of the newly added layer of 300 units for a given image was the feature representation associated with the image referent of the word, with the 10 vector representations averaged. The feature extraction was done trial-wise accordingly for each participant. We selected the above three models on the following grounds. Evidence[5] has shown that the DenseNet169 [49,50] can explain on average more variance of the brain response than many other computer vision models. MobileNetV2 was chosen to represent a simpler computer vision model that had less parameters (VGG19 has 143 667 240, DenseNet169 has 14 307 880 and MobileNetV2 has 3 538 984). VGG19 was chosen to represent a baseline model that had shallower architecture (VGG19 has 26 layers, DenseNet169 has 169 layers and MobileNetV2 has 88 layers), which has been well studied by previous studies [51–53].

To visualize the representational patterns of the computer vision models, we computed the representational dissimilarity matrices (RDMs) of the models similarly to the word embedding models. These are shown in the electronic supplementary material (DenseNet169, figure S44; MobileNetV2, figure S5 and VGG19, figure S6).

### 2.6.7. Encoding model pipeline

The encoding model pipeline was the same as in Miyawaki *et al.* [54] and implemented in Nilearn [30,55]. After standardizing the feature representations by subtracting the mean and dividing by the standard deviation, the feature representations were mapped to the BOLD signal of a given ROI by means of L2 regularized regression (ridge regression) with the regularization term equal to 100, following Miyawaki *et al.*

To estimate the performance of the regression, we partitioned the data into 300 folds to perform cross-validation by stratified random shuffling similar to the decoding pipeline above (§2.6.2). It is important to note that labels of the BOLD signals were only used for cross-validation partitioning, but they were not used in the encoding model fitting nor testing procedure. In each fold, we randomly held out 20% of the data for testing, while the remaining 80% was used to fit the ridge regression model, using the feature representations from word embedding models or computer vision models as features and the BOLD signals as targets. Predictions were then derived for the held-out data. The proportion of variance explained in each voxel was computed for the predictions. An average estimate of the variance explained was calculated. The best possible score is 1. The score can be also negative if the model is worse than random guessing. Voxels that had positive variance-explained values were identified for further analysis [54,56] for each participant, ROI and condition. To estimate the empirical chance level performance of the encoding models, a random shuffling was added to the training phase during cross-validation before model fitting. The random shuffling was applied to the order of the samples for the features in the encoding models while the order of the samples for the targets remained the same.

# 3. Results

## 3.1. Behavioural results

To ensure that participants were attending to the items during the task, a few catch trials were randomly presented at different points in each session. These trials showed number words and required a response via button press. The total number of catch trials was kept the same in both shallow and deep processing sessions. Catch trial data from six initial participants could not be obtained due to a technical issue. The proportion of correct responses on catch trials was $0.80 \pm 0.28$ in the shallow processing, and $0.78 \pm 0.37$ in deep processing conditions, which did not differ ($t_{20} = -0.40$, $p = 0.69$), hence showing that participants were equally engaged with the task in both conditions.

## 3.2. Multivariate classification analyses: decoding results

For the classification analyses, we used two types of cross-validation. In the first type, independent scans for training and testing were used (i.e. training occurred with partitions containing 80% of the

---

[5]http://www.brain-score.org/

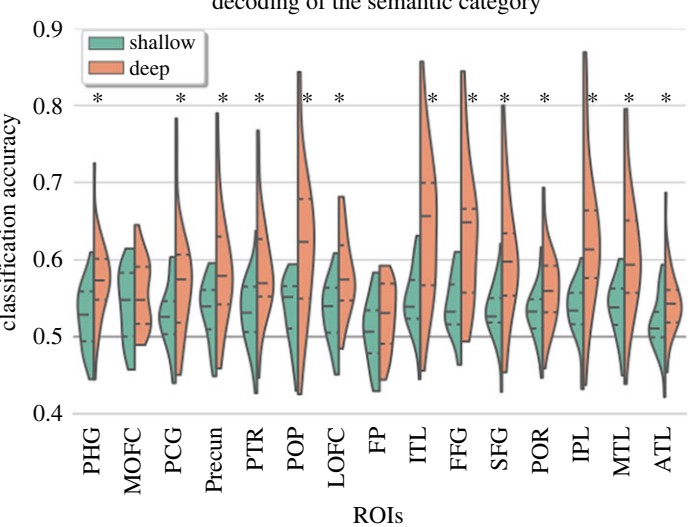

**Figure 3.** The figure shows summary statistics of decoding accuracy for the different ROIs. The three dotted lines inside each violin are the quartiles. The black asterisks mark ROIs that showed statistically significant improvement in decoding accuracy in deep as compared with shallow processing condition (FDR corrected for multiple comparisons).

data and testing with the remaining 20% and 300 sets of balanced train-test partitions were used), however, the scans could both refer to the same words. In the second type of cross-validation, the training and testing partitions did not refer to similar words (i.e. scans associated with a pair of words—one living and one non-living—was left out for testing) hence involving out-of-sample generalization.

Figure 3 presents the summary statistics of decoding accuracy using the first cross-validation approach, across ROIs in both shallow and deep processing conditions (i.e. during mental simulation). Notably, decoding in many regions of the semantic network was higher during deep processing compared with the shallow processing condition (figure 3). During shallow processing, decoding accuracy of the semantic category (living/non-living) was significantly above chance in most ROIs except the frontal pole ($50.74 \pm 4.38$, $t_{26} = 0.87$, $p = 0.39$): fusiform gyrus ($54.17 \pm 3.95$, $t_{26} = 5.39$, $p = 0.00009$), inferior parietal lobe ($53.20 \pm 3.75$, $t_{26} = 4.34$, $p = 0.0003$), inferior temporal lobe ($54.70 \pm 4.19$, $t_{26} = 5.71$, $p = 0.00008$), lateral orbitofrontal cortex ($53.42 \pm 3.96$, $t_{26} = 4.40$, $p = 0.0003$), medial orbitofrontal cortex ($54.23 \pm 4.65$, $t_{26} = 4.64$, $p = 0.0003$), middle temporal lobe ($53.79 \pm 3.78$, $t_{26} = 5.10$, $p = 0.0001$), pars opercularis ($53.51 \pm 4.27$, $t_{26} = 4.18$, $p = 0.0004$), pars orbitalis ($53.11 \pm 3.59$, $t_{26} = 4.42$, $p = 0.0003$), pars triangularis ($53.22 \pm 4.48$, $t_{26} = 3.66$, $p = 0.001$), parahippocampal gyrus ($52.63 \pm 4.11$, $t_{26} = 3.25$, $p = 0.004$), post-cingulate gyrus ($52.78 \pm 4.17$, $t_{26} = 3.39$, $p = 0.003$), precuneus ($53.61 \pm 3.95$, $t_{26} = 4.66$, $p = 0.0003$), superior frontal lobe ($53.27 \pm 3.78$, $t_{26} = 4.41$, $p = 0.0003$) and temporal pole ($51.64 \pm 3.73$, $t_{26} = 2.25$, $p = 0.036$).

In the deep processing condition, decoding accuracy was significantly above chance in all pre-specified ROIs (frontal pole ($52.62 \pm 4.36$, $t_{26} = 3.07$, $p = 0.005$), fusiform gyrus ($63.75 \pm 9.22$, $t_{26} = 7.61$, $p = 2.23 \times 10^{-7}$), inferior parietal lobe ($63.31 \pm 9.72$, $t_{26} = 6.98$, $p = 5.93 \times 10^{-7}$), inferior temporal lobe ($64.18 \pm 9.50$, $t_{26} = 7.61$, $p = 2.23 \times 10^{-7}$), lateral orbitofrontal cortex ($58.12 \pm 5.19$, $t_{26} = 7.98$, $p = 2.23 \times 10^{-7}$), medial orbitofrontal cortex ($55.60 \pm 4.38$, $t_{26} = 6.52$, $p = 1.19 \times 10^{-6}$), middle temporal lobe ($60.97 \pm 8.08$, $t_{26} = 6.92$, $p = 5.93 \times 10^{-7}$), pars opercularis ($61.92 \pm 9.65$, $t_{26} = 6.30$, $p = 1.70 \times 10^{-6}$), pars orbitalis ($56.15 \pm 4.84$, $t_{26} = 6.49$, $p = 1.19 \times 10^{-6}$), pars triangularis ($58.68 \pm 4.48$, $t_{26} = 6.68$, $p = 9.45 \times 10^{-7}$), parahippocampal gyrus ($57.30 \pm 5.23$, $t_{26} = 7.12$, $p = 5.47e-07$), post-cingulate gyrus ($57.06 \pm 7.24$, $t_{26} = 4.97$, $p = 3.88 \times 10^{-5}$), precuneus ($58.55 \pm 7.41$, $t_{26} = 5.88$, $p = 4.16 \times 10^{-6}$), superior frontal lobe ($60.10 \pm 8.23$, $t_{26} = 6.26$, $p = 1.74 \times 10^{-6}$) and temporal pole ($54.51 \pm 4.44$, $t_{26} = 5.19$, $p = 0.2.36 \times 10^{-5}$)).

We also repeated the analyses using a leave-one-run-out cross-validation procedure. Here we found significant decoding accuracy in the deep processing condition (see electronic supplementary material, figure S7). However, classification accuracy was at chance in all ROIs in the shallow processing case. Some caution, however, has to be taken here because there are only four cross-validation folds. It is known that leave-one-out procedures can lead to unstable estimates due to across-fold variability, and repeated random splits procedures like the one used above are preferred [57].

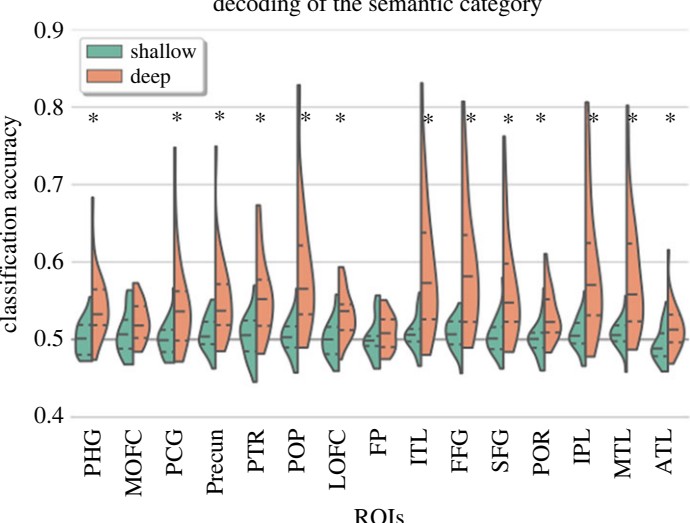

**Figure 4.** The figure shows summary statistics of the ROIs for out-of-sample generalization of the semantic category. The three dotted lines inside each violin are the quartiles. The black asterisks mark ROIs that showed statistically significant improvement in decoding accuracy in deep as compared with shallow processing condition (FDR corrected for multiple comparisons). PHG, parahippocampal gyrus; MOFC, medial orbitofrontal cortex; PCG, posterior cingulate gyrus; Precun, precuneus; PTR, pars triangularis; POP, pars opercularis; LOFC, lateral orbitofrontal cortex; FP, frontal pole; ITL, inferior temporal lobe; FFG, fusiform gyrus; SFG, superior frontal gyrus; POR, pars orbitalis; IPL, Inferior parietal lobe; MTL, middle temporal lobe; ATL, anterior temporal lobe.

## 3.3. Out-of-sample generalization

We then repeated the decoding analyses with a different cross-validation procedure that allowed testing the generalizability of the semantic representations and how this was modulated by the different task contexts. Specifically, the classifier was trained using all the words but leaving a pair of words out from each class. Then the classifier was tested on the left-out pair.

Figure 4 presents the summary statistics of the ROIs for out-of-sample generalization in both shallow and deep processing conditions. It can be seen that in the shallow processing condition, the decoding of the semantic category (living/non-living) was found to be at chance-level in all pre-specified ROIs including the frontal pole ($50.23 \pm 2.60$, $t_{26} = 0.46$, $p = 0.82$), fusiform gyrus ($50.70 \pm 2.02$, $t_{26} = 1.76$, $p = 0.28$), inferior parietal lobe ($50.76 \pm 2.15$, $t_{26} = 1.80$, $p = 0.27$), inferior temporal lobe ($50.76 \pm 2.13$, $t_{26} = 1.81$, $p = 0.27$), lateral orbitofrontal cortex ($50.17 \pm 2.44$, $t_{26} = 0.36$, $p = 0.82$), medial orbitofrontal cortex ($50.92 \pm 2.85$, $t_{26} = 1.64$, $p = 0.28$), middle temporal lobe ($50.74 \pm 1.91$, $t_{26} = 1.97$, $p = 0.27$), pars opercularis ($50.44 \pm 2.36$, $t_{26} = 0.95$, $p = 0.59$), pars orbitalis ($50.10 \pm 2.25$, $t_{26} = 0.23$, $p = 0.82$), pars triangularis ($50.46 \pm 2.89$, $t_{26} = 0.82$, $p = 0.63$), parahippocampal gyrus ($50.26 \pm 2.20$, $t_{26} = 0.59$, $p = 0.76$), post-cingulate gyrus ($50.13 \pm 2.23$, $t_{26} = 0.30$, $p = 0.82$), precuneus ($50.76 \pm 2.18$, $t_{26} = 1.77$, $p = 0.27$), superior frontal lobe ($50.50 \pm 2.33$, $t_{26} = 1.09$, $p = 0.54$) and temporal pole ($49.36 \pm 2.21$, $t_{26} = -1.48$, $p = 0.32$).

On the other hand, in the deep processing condition, the classification of the semantic category was found to be significantly above chance in all pre-specified ROIs (frontal pole ($50.98 \pm 2.24$, $t_{26} = 2.24$, $p = 0.03$), fusiform gyrus ($59.43 \pm 8.06$, $t_{26} = 5.97$, $p = 2.10 \times 10^{-5}$), inferior parietal lobe ($59.25 \pm 8.64$, $t_{26} = 5.45$, $p = 2.10 \times 10^{-5}$), inferior temporal lobe ($59.17 \pm 8.35$, $t_{26} = 5.60$, $p = 2.10 \times 10^{-5}$), lateral orbitofrontal cortex ($53.30 \pm 2.92$, $t_{26} = 5.78$, $p = 2.10 \times 10^{-5}$), medial orbitofrontal cortex ($52.34 \pm 2.50$, $t_{26} = 4.77$, $p = 7.60 \times 10^{-5}$), middle temporal lobe ($58.29 \pm 7.50$, $t_{26} = 5.64$, $p = 2.10 \times 10^{-5}$), pars opercularis ($59.27 \pm 8.60$, $t_{26} = 5.50$, $p = 2.10 \times 10^{-5}$), pars orbitalis ($53.18 \pm 3.13$, $t_{26} = 5.19$, $p = 3.40 \times 10^{-5}$), pars triangularis ($55.78 \pm 5.09$, $t_{26} = 5.79$, $p = 2.10 \times 10^{-5}$), parahippocampal gyrus ($54.50 \pm 4.22$, $t_{26} = 5.43$, $p = 2.10 \times 10^{-5}$), post-cingulate gyrus ($54.59 \pm 6.04$, $t_{26} = 3.88$, $p = 0.0007$), precuneus ($55.22 \pm 5.40$ $t_{26} = 4.93$, $p = 5.5 \times 10^{-5}$), superior frontal lobe ($56.67 \pm 6.83$, $t_{26} = 4.98$, $p = 5.3 \times 10^{-5}$) and temporal pole ($51.57 \pm 2.97$, $t_{26} = 2.68$, $p = 0.01$).

Decoding accuracy in 13 out of 15 ROIs was also better than in the shallow processing condition (figure 4). This shows that generalization of brain representations of semantic knowledge is better when the depth of processing is higher during mental simulation.

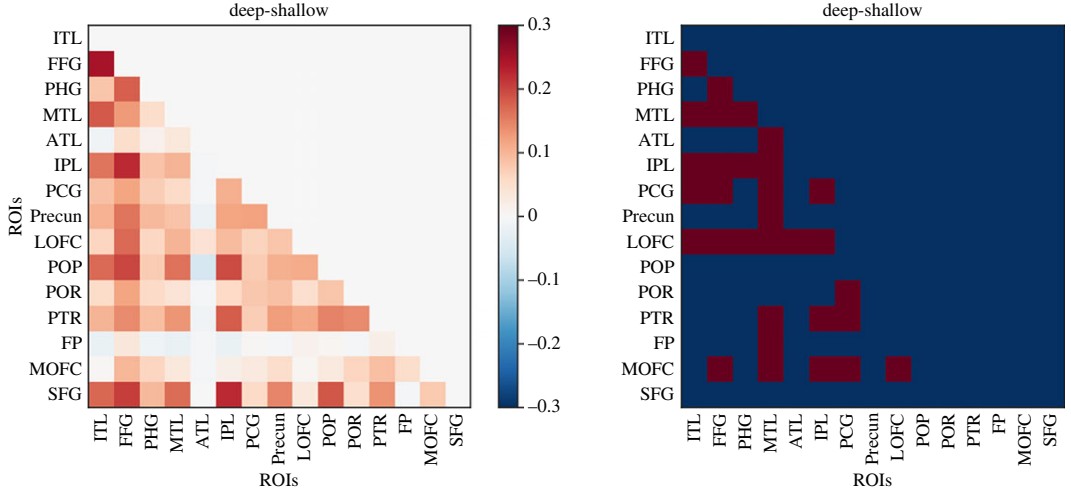

**Figure 5.** The figure shows the mean difference of informational connectivity between deep and shallow conditions. The right panel shows which of the pairs of ROIs were significantly more connected in deep as compared with the shallow condition (FDR corrected for multiple comparisons).

## 3.4. Informational connectivity results

We then assessed changes in the informational connectivity across the different ROIs, during deep relative to shallow processing conditions. This approach assesses the temporal correlation of multivariate patterns of responses across different ROIs and can reveal whether different brain regions interact in terms of the specific information that they carry [37].

Figure 5 illustrates the regions showing statistically higher correlations of decoding time courses during mental simulation compared with the shallow processing. There were two substrates that showed the strongest changes in informational connectivity with the rest of the semantic network. First, informational connectivity between the left orbitofrontal cortex increased with inferior temporal, parahippocampal, fusiform, inferior temporal, anterior temporal lobe and inferior parietal. A similar effect involved the middle temporal cortex, which showed increased informational connectivity during mental simulation with pars triangularis, medial and lateral orbitofrontal, frontal pole, precuneus and posterior cingulate and anterior temporal lobe. Figure 5 also shows that the pars opercularis, and superior frontal lobe showed no changes in informational connectivity as function of the depth of processing.

There was only one pair of ROIs in which the level of informational connectivity was higher in the shallow compared with deep processing condition, i.e. frontal pole and middle temporal lobe.

It may be argued that the differences in information connectivity may be explained by differences in the temporal correlation between the time courses of the different regions (i.e. the functional connectivity). We argue that if this were the case then we would expect to find the same pattern of differences across conditions at the level of functional connectivity too. To address this concern, we performed functional connectivity analysis as follows. First, the fMRI data were preprocessed and prepared as presented in §§2.4 and 2.6.1. Next, a time series was obtained for each ROI by taking the mean of all its voxels across each of the scans. To calculate the functional connectivity between the ROIs, these time series were correlated using Pearson correlation for each of the pairs of ROIs, and a matrix of functional connectivity was created. This procedure was performed separately for shallow and deep processing sessions resulting in two matrices for each of the participants. Finally, to compare between the functional connectivity in deep and shallow conditions, a paired *t*-test was conducted with FDR correction. The results are presented in the electronic supplementary material, figure S8, where it can be seen that there were no pairs of ROIs for which the functional connectivity was found to be significantly different across the deep and shallow conditions. This shows that the pattern of informational connectivity between ROIs cannot be explained by the functional connectivity between them.

Finally, it could be argued that the overall high decoding performance in the deep condition may be inflating the corresponding informational connectivity score matrices. We conducted the following analysis to address this. For each pair of ROIs, we defined the 'ground truth' value of Pearson's correlation, namely, the correlation between the unshuffled MVP discriminability vectors of two given

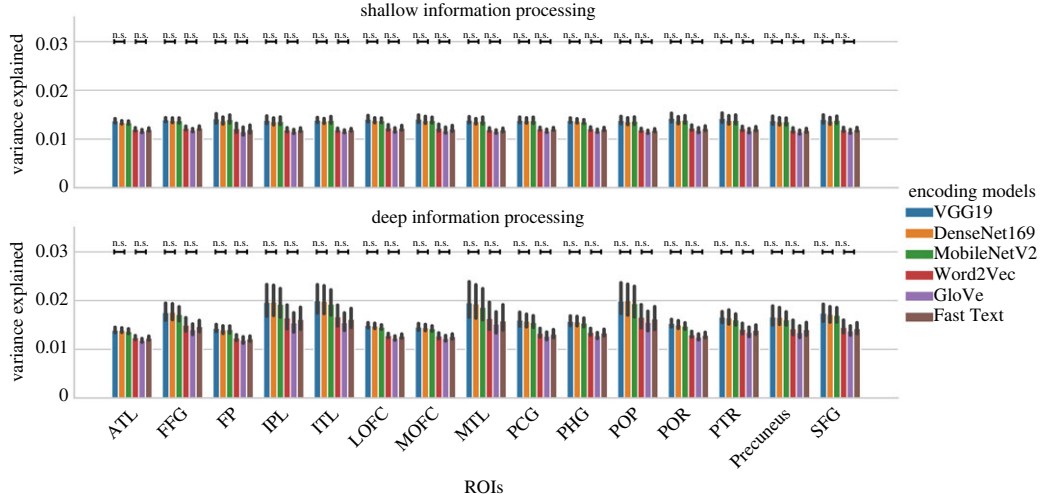

**Figure 6.** Average variance explained by each of the word embedding and computer vision models.

ROIs. Next, we randomly shuffled the first ROI's discriminability vector 10 000 times, and each time we correlated the shuffled vector to the second ROI's unshuffled vector, and thus created a null distribution of correlation values. Next, we computed the difference between the ground-truth and the mean of the null distribution and divided it by the variance of the distribution, thus calculating a measure of effect size for each pair of ROIs. Note this is independent of the level of classification performance. To compare this effect size between deep and shallow conditions, paired $t$-tests with FDR correction were run across subjects. We found that for all pairs of ROIs, this effect size was statistically significantly greater in the deep compared with the shallow condition, except for the pair involving the precuneus and superior frontal gyrus ($p = 0.053$). This shows that the higher level of decoding accuracy in the deep processing condition cannot account for the informational connectivity results.

## 3.5. Encoding results

An encoding model predicts the brain activity patterns using a set of features that are (non)linearly transformed from the stimuli [58,59]. Word embedding models (i.e. Word2Vec-2013 [60]) provide a way to characterize the geometry of semantic spaces. Computer vision models (i.e. deep convolutional neural networks [61]) can also reveal the structural organization of meaning in the ventral visual pathway [39]. To examine the properties of the brain representations during word recognition, we tested both word embedding and also computer vision models based on the image referents of the words used in the experiment (see §§2.6.5 and 2.6.6).

Figure 6 shows the average variance explained by the computer vision (VGG19, Densent169, MobilenetV2) and the word embedding (Fast Text, GloVe, Word2Vec) models, averaging across 27 subjects. The error-bars represent 95% confidence interval of a bootstrapping of 1000 iterations. For each ROI, a one-way analysis of variance (ANOVA) was performed within the computer vision models and within the embedding models. The ANOVAs aimed to detect the difference in variance explained within a type of models. After all the ANOVAs were performed, FDR correction procedures were applied to the raw $p$-values to correct for the multiple comparison within each condition (deep versus shallow). There was no difference among the different word embedding models, and no difference among the computer vision models.

We then computed the difference between each of the computer vision models and each of the word embedding models within each ROI and condition, in order to assess whether word embedding models or computer vision models were better. One-sample $t$-tests against zero were conducted for each pair with FDR correction. All the computer vision models performed better than any of the word embedding models (figure 7).

Finally, we computed the mean of the differences in variance explained between the computer vision models and the word embedding models and then assessed the extent of this difference in the deep and shallow processing conditions by using paired $t$-tests with FDR correction. Figure 8 illustrates the pattern of results within each ROI. We found that the advantage of computer vision models over word embedding models was higher in the deep processing condition relative to the shallow processing in the FFG, IPL, ITL, PCG, POP and POR (figure 8).

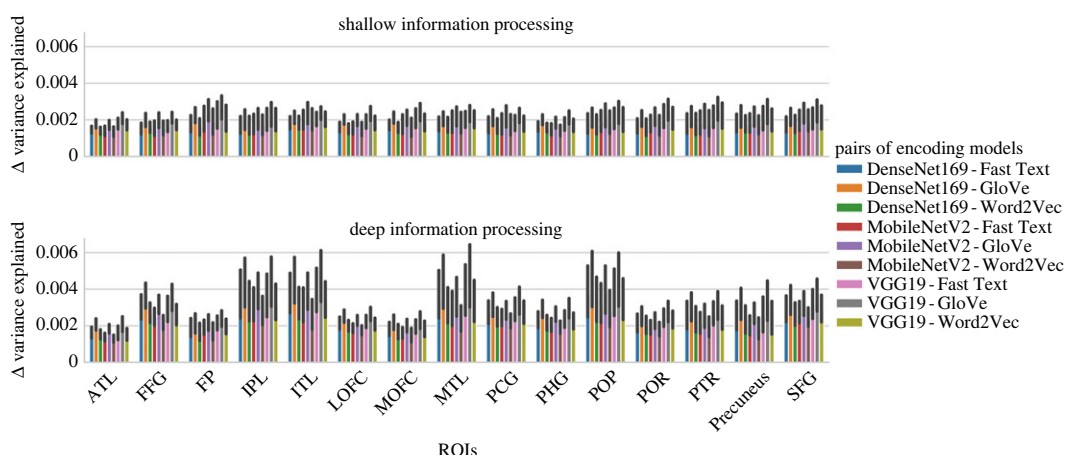

**Figure 7.** Differences between computer vision and word embedding models in variance explained. Computer vision models significantly explained more variance of the BOLD response compared with word embedding models. All one-sample *t*-tests against zero difference between models were significant and FDR corrected for multiple comparisons.

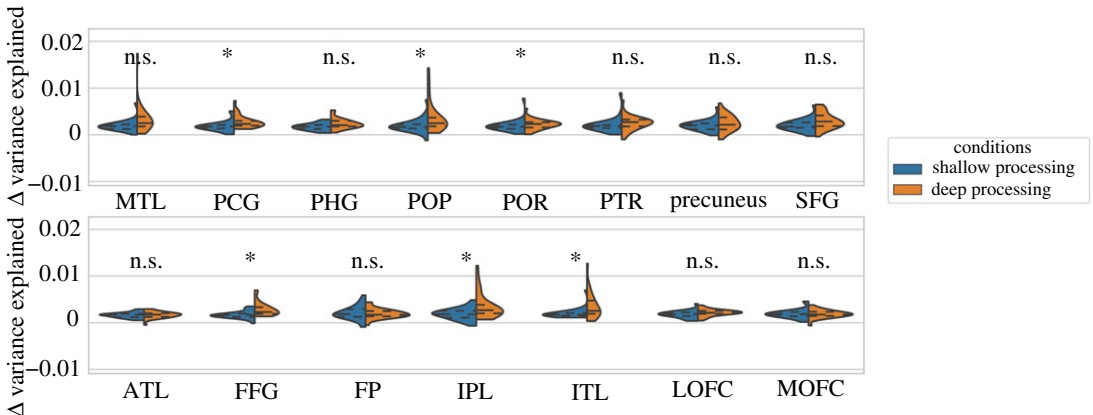

**Figure 8.** Overall difference between word embedding and computer vision models per ROI (*FDR corrected for multiple comparisons).

Note the above results were obtained with pre-trained computer vision models that were fine-tuned to ensure that the abstract layer had 300-D for comparison with the word embedding models. Otherwise, the results might not be comparable due to differences in the dimensionality of the models. However, we performed the same encoding analyses using the original dimensionality of the computer vision models and similar patterns were obtained. This indicates the robustness of the encoding results (see electronic supplementary material, figures S9–S11).

It may be argued that since only voxels with positive variance were identified for further analysis [54,56] there is a potential bias in this analysis. However, further analyses revealed that computer vision models generally explained more voxels with positive variance than the word embedding models (electronic supplementary material, figure S12). The difference between the word embedding and computer vision models in the number of voxels with positive variance explained was statistically significant (electronic supplementary material, figure S13), showing that computer vision models were better (one sample permutation *t*-tests, FDR-corrected for multiple comparisons). Additionally, if a single voxel was positively explained by the word embedding models, it was more likely that the very same voxel was better explained by the computer vision models (electronic supplementary material, figures S14 and S15).

## 4. Discussion

We sought to understand how the depth of processing shapes the brain representations of conceptual knowledge by using both decoding and encoding models. The results clearly show that the decoding

of word category information in most of the putative substrates of the semantic network [29] is consistently higher during the mental simulation condition relative to a shallow processing condition in which participants merely read the words. Semantic category was also decoded in the read condition, though decoding performance was far lower by comparison and the classifier did not reliably generalize to new examples that were not seen during training. By contrast, the generalizability of the classifier at predicting the semantic category of out-of-sample examples increased during mental simulation. Shallower processing modes are sufficient for in-sample classification but not enough for out-of-sample generalization (see also [62]).

Significant decoding of semantic category was observed in multiple areas of association (transmodal) cortex involving the middle temporal gyrus, anterior and inferior temporal, inferior parietal lobe and prefrontal substrates. Of particular relevance are the prefrontal areas, which are typically thought to be involved in semantic control, rather than representing semantic knowledge [63–65]. The present multivariate classification results are consistent with a role of association cortex in the representation of semantic categories too. Interestingly, the level of informational connectivity [66] observed during mental simulation also indicates that association transmodal cortices interact with multiple regions of the semantic network in terms of the specific information that multivoxel activity patterns carry across time. Relative to the shallow processing condition, informational connectivity increased during mental simulation between the left anterior prefrontal cortex, inferior parietal, temporal and occipital areas. Likewise, the middle temporal cortex—a region implicated in semantic control according to a prior meta-analyses [67]—showed increased informational connectivity with inferior frontal and anterior prefrontal areas, the posterior cingulate and anterior temporal lobe. We propose that the depth of information processing associated with mental simulation involved the broadcasting of the information across a distributed set of areas of the semantic network, hence making information globally available and consciously accessible. The global availability of information across the brain networks involved in semantic processing may be critical for the successful retrieval and manipulation of conceptual knowledge to guide thought and behaviour.

Encoding models further revealed the properties of the semantic representations encoded in brain activity across the different processing depths. We found that computer vision models outperformed word embedding models in explaining brain responses across the different regions of the semantic network. We found that the embedding layer of the computer vision model, which is likely to contain a condensed summarization of visual features that are frequent in the examples of the image referents for a given word, explained more variance of the brain responses. We found that the advantage of computer vision models over word embedding models was higher during mental simulation—in the deep compared with the shallow processing condition. In particular, during deep processing computer vision models explained more variance in areas of the ventral visual pathway, including the fusiform, inferior temporal and inferior parietal cortex, and also in the posterior cingulate and inferior frontal cortex. This suggests that access to visual representations is more likely when the depth of processing is higher during mental simulation. However, the advantage of computer vision models over word embedding models occurred independently of the depth of processing in multiple areas including the middle temporal lobe, the parahippocampal gyrus, precuneus, superior frontal gyrus and anterior prefrontal cortex. This is, however, consistent with the view that conceptual knowledge can also rely on modality-specific representations that need no conscious access to visual (i.e. imagery-based) processing [68,69].

The encoding results indicate that the semantic and syntactic similarity metrics provided by word embedding models may not capture the actual semantic knowledge that is represented in the brain during word processing. Rather, it appears that image-based properties are necessary to account for how the brain represents word meaning. That this occurred both during mental simulation and also during the shallow processing condition is consistent with models that propose simulation processes may occur, to some extent, automatically [22,70,71]. However, it is clear from the superior decoding performance and the increased informational connectivity during mental simulation that top-down factors associated with the depth of processing also play a key modulating role. Together, these results demonstrate that the depth of processing during mental simulation is a key factor for triggering highly distributed, generalizable and integrated brain representations in the semantic network.

One limitation of the current study is that task manipulation was relatively coarse grained, i.e. mental simulation versus covert reading of the items. Our goal here was to examine how the depth of information processing influenced the brain representation of meaning and, accordingly, we elected to compare task contexts that differed in the need to consciously represent the meaning of the items. This is important because there has been little research on the role of task factors at influencing the

brain representation of meaning. Seminal studies on neural correlates of the levels of processing effect [72] used univariate approaches which are not well suited to understand the brain representation of meaning. The effect of mental simulation on classification accuracy is, however, unlikely to be due to differences in the degree of attentional allocation to the items and differences in task engagement. First, catch trials were present across the different task contexts and performance was similar, suggesting that participants deployed attention and were engaged with the tasks to a similar extent. Instead, we interpret the effects of mental simulation in terms of the re-enactment of the sensorimotor processes associated with the perception of the referents of the words. Nevertheless, we acknowledge that further studies tapping on more fine-grained manipulations of semantic processing (i.e. lexical versus concreteness decisions or feature versus category verification tasks) and sensory experience (i.e. visual versus auditory) including a wider range of stimulus classes, may be useful to pinpoint more precisely how the nature of the specific task shapes how the brain encodes meaning. Additional work needs to be conducted to make further determinations.

Ethics. The study conformed to the Declaration of Helsinki and was approved by the BCBL Research Ethics Board.
Data accessibility. Neuroimaging data and analyses scripts are available in the Dryad Digital Repository associated with this paper (see https://doi.org/10.5061/dryad.vmcvdncpf) [73].
Authors' contributions. D.S. and U.A.S. designed the study; D.S. and U.A.S. ran the experiments and performed the decoding analyses; U.A.S. developed the pipeline for the informational connectivity analysis; N.M. and R.S. were involved in testing the encoding models; all authors contributed towards the writing of the paper and approved the final version of the manuscript; D.S. supervised all aspects of the project.
Competing interests. The authors declare no competing interests.
Funding. D.S. acknowledges support from the Basque Government through the BERC 2018-2021 programme, from the Spanish Ministry of Economy and Competitiveness, through the 'Severo Ochoa' Programme for Centres/Units of Excellence in R&D (SEV-2015-490) and also from project grants PSI2016-76443-P from MINECO and PI-2017-25 from the Basque Government. R.S. acknowledges support by the Basque Government (IT1244-19 and ELKARTEK programmes), and the Spanish Ministry of Economy and Competitiveness MINECO (project TIN2016-78365-R).

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
