## [Reviewer comments · Royal Society Open Science]

Review History

RSOS-192043.R0 (Original submission)

Review form: Reviewer 1

Is the manuscript scientifically sound in its present form?

No

Are the interpretations and conclusions justified by the results?

Yes

Is the language acceptable?

Yes

Do you have any ethical concerns with this paper?

No

Have you any concerns about statistical analyses in this paper?

Yes

Recommendation?

Major revision is needed (please make suggestions in comments)

Comments to the Author(s)

In this work, the authors report an investigation of how conceptual knowledge is represented in the brain. They examine how depth of processing (modulated by task) affects the representation of semantic categories for visual words through an fMRI scan and subsequent analyses. Depth of processing (shallow versus deep) was varied by having participants read words, or mentally simulate the associated features. A series of analyses, including multivariate classification, informational connectivity, and encoding models were employed to compare the conditions. Deep processing increased decoding performance (relative to shallow processing) in regions of the semantic network, and informational connectivity was strengthened between semantic regions. Computer vision models, though not word embeddings, explained variance of the representations across multiple regions.

General comments

This study addresses the interesting question of how neural representations for semantic/conceptual knowledge are affected by current cognitive processing. The manuscript is well written (aside from some minor typos) and the study reports results from an impressive range of methods that are nicely complementary for the study's goals. There are several areas, however, that need to be adjusted before publication. The classification cross-validation procedure should leave out data from untrained runs, rather than timepoints of the same run. Additionally, the informational connectivity comparisons should account for differences in classification performance (see below for one way to do this). Finally, the manuscript would benefit from some further discussion of the difference between processing depth, and the cognitive target of processing. With these changes, the manuscript will make an interesting and useful contribution to the literature.

Specific comments

Major

1) The cross-validation framework holds out items in an 80-20% split (with 300 sets created by randomization). It would be better to hold-out runs as independent testing data. This is because items within the same run can have overlapping hemodynamic responses, so without holding out runs, the training and testing sets are not independent. Similarly, processing such as Z-scoring and exposure to noise from a run can influence a classifier to give above-chance performance. Because of the counterbalanced order of the blocks, it would be best to hold two runs out, to ensure the training set is balanced in this regard (see Snoek et al., 2019, NeuroImage).

2) The PCA used to select features (p. 8) should be conducted in the training data of each cross-validation fold (and then applied to the held-out data). This ensures the testing data does not influence the feature selection (and thus final classification performance).

3) How might the focus of cognitive processing influence the differences observed between deep and shallow conditions. For example, the process target of the shallow task is reading the words, versus imaging properties of the item. It seems that two differences are present in the conditions – between the depth of processing, and between the particular characteristics emphasized when encountering the words. This is somewhat unavoidable, but it would be worth discussing what this means for the conclusions about how depth processing modulates the representation of meaning, rather than reflecting differences in focus of attention, irrespective of 'depth' per se. For instance, a third equally deep condition (focusing on other characteristics) might in theory help with this.

4) The comparison of informational connectivity strengths between the two conditions is very interesting, but it is necessary to control for the significantly different levels of classification that are reported elsewhere. When one condition has superior classification (here, deep processing), it is more likely by chance that two regions will be more correlated (due to having more higher values in common). One way is to calculate statistical significance by permuting (scrambling) one region's values to create a null distribution of informational connectivity. This will ensure that the

overall decoding performance (or distance from the classifier's bounds) is not inflating the deep-processing's connectivity matrix, relative to the shallow-processing for the same basis of differences in overall performance.

5) Please list the scanning parameters. For example, orientation of slice acquisition, TR, voxel size, etc.

Minor

6) The data appears to be behind a request form.

7) It would be helpful to translate the words into English (page 5). This is completed in the figure, but not the list of words

8) Were all the stimuli known to participants? It would be worth reporting familiarity statistics from an available norming dataset, or an independent set of participants. This is important because it is naturally more difficult to simulate features for concepts that are not as familiar.

9) On page 20, the authors state that "The depth of processing associated with mental simulation therefore triggered a general broadcasting effect of the specific content being represented". What does "general broadcasting" mean here? Different terminology might make the implications clearer to the reader. Do you mean the signal is sent elsewhere? If so, why?

Review form: Reviewer 2

Is the manuscript scientifically sound in its present form?

No

Are the interpretations and conclusions justified by the results?

No

Is the language acceptable?

Yes

Do you have any ethical concerns with this paper?

No

Have you any concerns about statistical analyses in this paper?

No

Recommendation?

Major revision is needed (please make suggestions in comments)

Comments to the Author(s)

Review of "Decoding and encoding models reveal the role of the depth of processing in the brain representation of meaning"

Manuscript number: RSOS-192043

Authors: David Soto, Usman Ayub Sheikh, Ning Mei, and Roberto Santana

This study investigates how "depth of processing" affects the neural representation of semantic categories as measured through fMRI multi-voxel pattern analysis. Participants performed a visual word recognition task with two conditions: in the "shallow processing" condition participants were instructed to passively read the words, while in the "deep processing" condition they were instructed to "mentally simulate the properties associated with the word".

Conditions were blocked by scanning runs. The stimuli consisted of 36 concrete nouns (18 animal names and 18 tool names). Participants provided overt responses only to catch trials, in which a number word was presented.

Three different analyses were performed: (1) A support vector machine pattern classification analysis was used to assess the extent to which semantic category information (living/nonliving) was present in the fMRI activation patterns in each condition (“shallow” vs. “deep” processing). This analysis was performed separately in 15 pre-defined regions of interest (ROIs). It revealed that, in 13 of the 15 ROIs, classification accuracy was higher in the deep processing condition. (2) An informational connectivity analysis was used to characterize the temporal correlation of semantic category discriminability between ROIs over the course of the trial. The purpose of this analysis was to investigate whether this informational connectivity between ROIs varies across deep and shallow processing conditions. It revealed that the temporal correlation between regions was significantly higher in the deep processing condition for several pairs of ROIs. The authors do not report whether any pairs of ROIs were significantly more connected in shallow compared to deep processing conditions. (3) Finally, an encoding model analysis compared the proportion of variance explained by distributional word embedding models (Fast Text, GloVe, and Word2vec) and computer vision models (VGG19, MobileNetV2, and DenseNet169), separately for each ROI and each condition (shallow vs. deep processing). No differences were found between the three word embedding models or between the three computer vision models, but computer vision models performed significantly better than word embedding models. The effect of condition, however, was mixed: “the advantage of computer vision models over word embeddings models was higher in the deep processing condition relative to the shallow processing in PCG, PHG, and POP, while the opposite pattern was observed in FFG, IPL, and ITL.”

The study addresses an important issue in the cognitive neuroscience of conceptual thinking and language processing, namely, the nature of the code underlying the neural representation of word meaning. However, I do have substantial concerns about its specific goals, methods, and interpretation of results, which are listed below.

Major concerns:

1. The manuscript does not appropriately contextualize the study within the neuroimaging literature on concept representation (e.g., Devereux, B. J. et al., 2013. Representational similarity analysis reveals commonalities and differences in the semantic processing of words and objects. *The Journal of Neuroscience*, 33(48); Anderson, A. J. et al., 2015. Reading visually embodied meaning from the brain: Visually grounded computational models decode visual-object mental imagery induced by written text. *NeuroImage*, 120; Fernandino, L. et al., 2016. Heteromodal Cortical Areas Encode Sensory-Motor Features of Word Meaning. *Journal of Neuroscience*, 36(38); Martin, C. B. et al., 2018. Integrative and distinctive coding of visual and conceptual object features in the ventral visual stream. *eLife*, 7, 1493; Carota, F. et al., 2017. Representational Similarity Mapping of Distributional Semantics in Left Inferior Frontal, Middle Temporal, and Motor Cortex. *Cerebral Cortex*, 27(1); Devereux, B.J. et al., 2018. Integrated deep visual and semantic attractor neural networks predict fMRI pattern-information along the ventral object processing pathway. *Sci Rep* 8, 10636; Anderson, A. J. et al. (2019). An Integrated Neural Decoder of Linguistic and Experiential Meaning. *The Journal of Neuroscience*, 39(45)), levels of processing (e.g., Nyberg, L., 2002. Levels of processing: A view from functional brain imaging. *Memory*, 10(5)) or mental imagery (e.g., Daselaar, S. M. et al., 2010. Modality-specific and modality-independent components of the human imagery system. *NeuroImage*, 52(2); Huijbers, W. et al., 2011. Imagery and retrieval of auditory and visual information: neural correlates of successful and unsuccessful performance. *Neuropsychologia*, 49(7)). The Introduction only cites a few hand-picked studies that, in my opinion, are not as relevant to the current study as some of the references above.
2. There is a discrepancy between the wide scope of the study’s claims, as stated in the title, abstract, introduction, and discussion, and the narrow scope of the semantic manipulations that were actually employed. In particular, the assessment of semantic discriminability was restricted to the distinction between two categories (animals and tools). It is not clear that the effects of task

demands found for this particular categorical distinction can be generalized to other categorical distinctions, much less to “representational spaces of conceptual knowledge” (p. 3, l. 37) in general. I suggest the authors rephrase their claims to reflect the actual scope of the investigation.

3. While the study’s main goal is to investigate “the role of depth of processing in the brain representation of meaning”, the term “depth of processing” is never defined. In the Introduction, the authors cite three studies as examples of deep processing, all from the same group and using the same task: participants are presented with a written word denoting a concrete concept and, simultaneously, a line drawing of the same concept, and are instructed to think about salient properties of the concept. As examples of shallow processing, they cite a study by Simanova et al. (2012), in which subjects were presented with four types of stimuli (visual words, auditory words, colored photographs, and non-verbal natural sounds) and instructed to “judge whether each stimulus within a block was semantically consistent with the others”, and a study by Huth et al. (2016), in which participants listened to pre-recorded narrative stories. These examples do not provide a clear picture of what the authors mean by “depth of processing”. It is, therefore, difficult to evaluate their claim that “previous research has not determined the role of the depth of processing on the brain representation of concepts” (p. 3, l. 39).

In the present study, “depth of processing” is operationalized as the difference between (a) passively reading an isolated concrete noun and (b) engaging in “mental simulation” of the noun. Again, it is not clear what “mental simulation” refers to, since the actual instructions to participants are not reported in the manuscript, but if we assume that in this condition participants were instructed to focus on the sensory-motor properties of the concept named, then mental imagery is the process that is actually under investigation. It is far from clear, however, that the effects of this mental imagery manipulation on the semantic discriminability of the stimulus categories can be generalized to all processes (or manipulations) that could reasonably fall under the umbrella of “depth of processing”. The cognitive neuroscience literature on word processing contains a wide variety of semantic manipulations that can be construed as different levels of processing depth, such as phonological decision, lexical decision, familiarity rating, concreteness rating, feature verification, category membership judgment, synonym matching, etc. I suggest the authors provide a clearer definition of the phenomenon under investigation and exercise more restraint in extrapolating the implications of the results to phenomena or constructs that were not directly investigated.

4. The hypotheses investigated in the study (last paragraph of the Introduction) seem somewhat trivial given previous results in the literature. With respect to the first hypothesis, while I am not aware of any studies that have specifically tested whether deeper semantic processing of word stimuli results in higher decoding accuracy for semantic categories, it seems to me that this hypothesis must be true at some level. If processing is shallow enough, we expect that the semantic content of the stimuli will be minimally activated, and fMRI activation patterns associated with semantic categories will not be distinguishable from each other. It is useful to know, however, that the specific manipulation employed in the current study did result in higher decoding accuracies across all ROIs.

It was not clear to me how the second hypothesis (about generalizability) differs from the first. If decoding accuracy is assessed by predicting the category of items not used for training, how are i and ii different? The procedures as described in the Methods do not help clarify the distinction between item categorization and item generalizability. The third hypothesis also seems to logically follow from the first: if concept representations are distributed over different cortical areas (as several studies have indicated), the time course of the activation patterns should be correlated across these areas (because of the time course of the hemodynamic response), and this correlation will, of course, be modulated by the discriminability of the activation patterns. Finally, it is not clear from the Introduction how the fourth hypothesis relates to the main goal of the study regarding the role of depth of processing.

5. No information on MRI acquisition protocols are provided in the Methods section.

6. What were the exact instructions for the mental simulation condition? Were participants instructed to focus on sensory-motor properties (shape, color, size, etc.) or were they encouraged to think of any properties of their own choosing? Were they instructed to engage in mental imagery?

7. How were the 15 ROIs specified? The meta-analysis cited (Binder et al., 2009) lists 7 cortical regions, most of them with bilateral representation, but in the present study all ROIs are in the left hemisphere. Why were right hemisphere regions not included? Also, how were the ROIs defined in the MRI volumes? If an atlas was used, it should be specified in the Methods.
8. "A set of 15 left-lateralized ROIs was pre-specified (see Figure 2) based on a meta-analysis of the semantic system [12] and one anterior temporal lobe (ATL) due to its role as a 'semantic hub'" (p. 6 l. 50). Although there is considerable evidence for the involvement of the anterior temporal lobe in semantic cognition, its role as a semantic hub has not yet been conclusively established.
9. Encoding model pipeline: "The proportion of variance explained in each voxel was computed for the predictions. An average estimate of the variance explained was calculated. The best possible score is 1. The score can be also negative if the model is worse than random guessing. Voxels that had positive variance explained values were identified for further analysis [42, 45] for each participant, ROI and condition." (p. 11, l. 42). It seems to me that this procedure would bias the results toward models with higher variance (across voxels) of proportion variance explained. Suppose, for instance, that Model A and Model B have the same average score across all voxels in a ROI, with Model A having higher positive scores and lower negative scores (i.e., higher variance of scores). If one looks only at the voxels with positive scores, Model A will erroneously appear to have a higher average score than B.
10. How many catch trials were presented in each block/run?
11. "Out-of-sample Generalization: We then repeated the decoding analyses with a different cross-validation procedure that allowed testing the generalizability of the semantic representations and how this was modulated by the different task contexts. Specifically, the classifier was trained using all the words but leaving a pair of words out from each class. Then the classifier was tested on the left-out pair." (p. 14, l. 10). It is not clear how this analysis evaluates out-of-sample generalization. How is it different from the previous decoding analysis?
12. It is not clear how the results of the informational connectivity analysis should be interpreted. Section 3.4 and Figure 5 describe a complex pattern of increases and decreases in temporal correlations between ROIs, but no attempt is made to explain this pattern or to argue how it is relevant to the goals of the study. On a related note, were any pairs of ROIs significantly more connected in shallow compared to deep processing conditions?
13. Is it possible that the difference in performance between computer vision models and word embedding models reflects differences in the model architectures (e.g., deep learning vs. shallow models) or in their training protocols rather than something about the nature of the neural representations themselves?
14. "Interestingly, the level of informational connectivity [53] observed during mental simulation also indicates that association transmodal cortices interact with multiple regions of the semantic network in terms of the specific information that multivoxel activity patterns carry across time." As I mentioned in point #4 above, it seems to me that the temporal correlations between areas could be completely explained by the time course of the hemodynamic response, in which case they would not reflect information connectivity.

Decision letter (RSOS-192043.R0)

27-Jan-2020

Dear Professor Soto,

The editors assigned to your paper ("Decoding and encoding models reveal the role of the depth of processing in the brain representation of meaning") have now received comments from reviewers. We would like you to revise your paper in accordance with the referee and Associate Editor suggestions which can be found below (not including confidential reports to the Editor). Please note this decision does not guarantee eventual acceptance.

Please submit a copy of your revised paper before 19-Feb-2020. Please note that the revision deadline will expire at 00.00am on this date. If we do not hear from you within this time then it will be assumed that the paper has been withdrawn. In exceptional circumstances, extensions may be possible if agreed with the Editorial Office in advance. We do not allow multiple rounds of revision so we urge you to make every effort to fully address all of the comments at this stage. If deemed necessary by the Editors, your manuscript will be sent back to one or more of the original reviewers for assessment. If the original reviewers are not available, we may invite new reviewers.

- Data accessibility

If you wish to submit your supporting data or code to Dryad (<http://datadryad.org/>), or modify your current submission to dryad, please use the following link:
<http://datadryad.org/submit?journalID=RSOS&manu=RSOS-192043>

- Competing interests

- Authors' contributions

- Acknowledgements

- Funding statement

on behalf of Dr César Lima (Associate Editor) and Essi Viding (Subject Editor)
openscience@royalsociety.org

Comments to Author:

Reviewers' Comments to Author:

Reviewer: 1

Comments to the Author(s)

In this work, the authors report an investigation of how conceptual knowledge is represented in the brain. They examine how depth of processing (modulated by task) affects the representation of semantic categories for visual words through an fMRI scan and subsequent analyses. Depth of processing (shallow versus deep) was varied by having participants read words, or mentally simulate the associated features. A series of analyses, including multivariate classification, informational connectivity, and encoding models were employed to compare the conditions. Deep processing increased decoding performance (relative to shallow processing) in regions of the semantic network, and informational connectivity was strengthened between semantic regions. Computer vision models, though not word embeddings, explained variance of the representations across multiple regions.

General comments

This study addresses the interesting question of how neural representations for semantic/conceptual knowledge are affected by current cognitive processing. The manuscript is well written (aside from some minor typos) and the study reports results from an impressive range of methods that are nicely complementary for the study's goals. There are several areas, however, that need to be adjusted before publication. The classification cross-validation procedure should leave out data from untrained runs, rather than timepoints of the same run.

Additionally, the informational connectivity comparisons should account for differences in classification performance (see below for one way to do this). Finally, the manuscript would benefit from some further discussion of the difference between processing depth, and the cognitive target of processing. With these changes, the manuscript will make an interesting and useful contribution to the literature.

Specific comments

Major

1) The cross-validation framework holds out items in an 80-20% split (with 300 sets created by randomization). It would be better to hold-out runs as independent testing data. This is because items within the same run can have overlapping hemodynamic responses, so without holding out runs, the training and testing sets are not independent. Similarly, processing such as Z-scoring and exposure to noise from a run can influence a classifier to give above-chance performance. Because of the counterbalanced order of the blocks, it would be best to hold two runs out, to ensure the training set is balanced in this regard (see Snoek et al., 2019, NeuroImage).

2) The PCA used to select features (p. 8) should be conducted in the training data of each cross-validation fold (and then applied to the held-out data). This ensures the testing data does not influence the feature selection (and thus final classification performance).

3) How might the focus of cognitive processing influence the differences observed between deep and shallow conditions. For example, the process target of the shallow task is reading the words, versus imaging properties of the item. It seems that two differences are present in the conditions – between the depth of processing, and between the particular characteristics emphasized when encountering the words. This is somewhat unavoidable, but it would be worth discussing what this means for the conclusions about how depth processing modulates the representation of meaning, rather than reflecting differences in focus of attention, irrespective of ‘depth’ per se. For instance, a third equally deep condition (focusing on other characteristics) might in theory help with this.

4) The comparison of informational connectivity strengths between the two conditions is very interesting, but it is necessary to control for the significantly different levels of classification that are reported elsewhere. When one condition has superior classification (here, deep processing), it is more likely by chance that two regions will be more correlated (due to having more higher values in common). One way is to calculate statistical significance by permuting (scrambling) one region’s values to create a null distribution of informational connectivity. This will ensure that the overall decoding performance (or distance from the classifier’s bounds) is not inflating the deep-processing’s connectivity matrix, relative to the shallow-processing for the same basis of differences in overall performance.

5) Please list the scanning parameters. For example, orientation of slice acquisition, TR, voxel size, etc.

Minor

6) The data appears to be behind a request form.

7) It would be helpful to translate the words into English (page 5). This is completed in the figure, but not the list of words

8) Were all the stimuli known to participants? It would be worth reporting familiarity statistics from an available norming dataset, or an independent set of participants. This is important because it is naturally more difficult to simulate features for concepts that are not as familiar.

9) On page 20, the authors state that “The depth of processing associated with mental simulation therefore triggered a general broadcasting effect of the specific content being represented”. What

does “general broadcasting” mean here? Different terminology might make the implications clearer to the reader. Do you mean the signal is sent elsewhere? If so, why?

Reviewer: 2

Comments to the Author(s)

Review of “Decoding and encoding models reveal the role of the depth of processing in the brain representation of meaning”

Manuscript number: RSOS-192043

Authors: David Soto, Usman Ayub Sheikh, Ning Mei, and Roberto Santana

This study investigates how “depth of processing” affects the neural representation of semantic categories as measured through fMRI multi-voxel pattern analysis. Participants performed a visual word recognition task with two conditions: in the “shallow processing” condition participants were instructed to passively read the words, while in the “deep processing” condition they were instructed to “mentally simulate the properties associated with the word”. Conditions were blocked by scanning runs. The stimuli consisted of 36 concrete nouns (18 animal names and 18 tool names). Participants provided overt responses only to catch trials, in which a number word was presented.

Three different analyses were performed: (1) A support vector machine pattern classification analysis was used to assess the extent to which semantic category information (living/nonliving) was present in the fMRI activation patterns in each condition (“shallow” vs. “deep” processing). This analysis was performed separately in 15 pre-defined regions of interest (ROIs). It revealed that, in 13 of the 15 ROIs, classification accuracy was higher in the deep processing condition. (2) An informational connectivity analysis was used to characterize the temporal correlation of semantic category discriminability between ROIs over the course of the trial. The purpose of this analysis was to investigate whether this informational connectivity between ROIs varies across deep and shallow processing conditions. It revealed that the temporal correlation between regions was significantly higher in the deep processing condition for several pairs of ROIs. The authors do not report whether any pairs of ROIs were significantly more connected in shallow compared to deep processing conditions. (3) Finally, an encoding model analysis compared the proportion of variance explained by distributional word embedding models (Fast Text, GloVe, and Word2vec) and computer vision models (VGG19, MobileNetV2, and DenseNet169), separately for each ROI and each condition (shallow vs. deep processing). No differences were found between the three word embedding models or between the three computer vision models, but computer vision models performed significantly better than word embedding models. The effect of condition, however, was mixed: “the advantage of computer vision models over word embeddings models was higher in the deep processing condition relative to the shallow processing in PCG, PHG, and POP, while the opposite pattern was observed in FFG, IPL, and ITL.”

The study addresses an important issue in the cognitive neuroscience of conceptual thinking and language processing, namely, the nature of the code underlying the neural representation of word meaning. However, I do have substantial concerns about its specific goals, methods, and interpretation of results, which are listed below.

Major concerns:

1. The manuscript does not appropriately contextualize the study within the neuroimaging literature on concept representation (e.g., Devereux, B. J. et al., 2013. Representational similarity analysis reveals commonalities and differences in the semantic processing of words and objects. *The Journal of Neuroscience*, 33(48); Anderson, A. J. et al., 2015. Reading visually embodied meaning from the brain: Visually grounded computational models decode visual-object mental imagery induced by written text. *NeuroImage*, 120; Fernandino, L. et al., 2016. Heteromodal Cortical Areas Encode Sensory-Motor Features of Word Meaning. *Journal of Neuroscience*, 36(38); Martin, C. B. et al., 2018. Integrative and distinctive coding of visual and conceptual object

features in the ventral visual stream. *eLife*, 7, 1493; Carota, F. et al., 2017. Representational Similarity Mapping of Distributional Semantics in Left Inferior Frontal, Middle Temporal, and Motor Cortex. *Cerebral Cortex*, 27(1); Devereux, B.J. et al., 2018. Integrated deep visual and semantic attractor neural networks predict fMRI pattern-information along the ventral object processing pathway. *Sci Rep* 8, 10636; Anderson, A. J. et al. (2019). An Integrated Neural Decoder of Linguistic and Experiential Meaning. *The Journal of Neuroscience*, 39(45)), levels of processing (e.g., Nyberg, L., 2002. Levels of processing: A view from functional brain imaging. *Memory*, 10(5)) or mental imagery (e.g., Daselaar, S. M. et al., 2010. Modality-specific and modality-independent components of the human imagery system. *NeuroImage*, 52(2); Huijbers, W. et al., 2011. Imagery and retrieval of auditory and visual information: neural correlates of successful and unsuccessful performance. *Neuropsychologia*, 49(7)). The Introduction only cites a few hand-picked studies that, in my opinion, are not as relevant to the current study as some of the references above.

2. There is a discrepancy between the wide scope of the study's claims, as stated in the title, abstract, introduction, and discussion, and the narrow scope of the semantic manipulations that were actually employed. In particular, the assessment of semantic discriminability was restricted to the distinction between two categories (animals and tools). It is not clear that the effects of task demands found for this particular categorical distinction can be generalized to other categorical distinctions, much less to "representational spaces of conceptual knowledge" (p. 3, l. 37) in general. I suggest the authors rephrase their claims to reflect the actual scope of the investigation.

3. While the study's main goal is to investigate "the role of depth of processing in the brain representation of meaning", the term "depth of processing" is never defined. In the Introduction, the authors cite three studies as examples of deep processing, all from the same group and using the same task: participants are presented with a written word denoting a concrete concept and, simultaneously, a line drawing of the same concept, and are instructed to think about salient properties of the concept. As examples of shallow processing, they cite a study by Simanova et al. (2012), in which subjects were presented with four types of stimuli (visual words, auditory words, colored photographs, and non-verbal natural sounds) and instructed to "judge whether each stimulus within a block was semantically consistent with the others", and a study by Huth et al. (2016), in which participants listened to pre-recorded narrative stories. These examples do not provide a clear picture of what the authors mean by "depth of processing". It is, therefore, difficult to evaluate their claim that "previous research has not determined the role of the depth of processing on the brain representation of concepts" (p. 3, l. 39).

In the present study, "depth of processing" is operationalized as the difference between (a) passively reading an isolated concrete noun and (b) engaging in "mental simulation" of the noun. Again, it is not clear what "mental simulation" refers to, since the actual instructions to participants are not reported in the manuscript, but if we assume that in this condition participants were instructed to focus on the sensory-motor properties of the concept named, then mental imagery is the process that is actually under investigation. It is far from clear, however, that the effects of this mental imagery manipulation on the semantic discriminability of the stimulus categories can be generalized to all processes (or manipulations) that could reasonably fall under the umbrella of "depth of processing". The cognitive neuroscience literature on word processing contains a wide variety of semantic manipulations that can be construed as different levels of processing depth, such as phonological decision, lexical decision, familiarity rating, concreteness rating, feature verification, category membership judgment, synonym matching, etc. I suggest the authors provide a clearer definition of the phenomenon under investigation and exercise more restraint in extrapolating the implications of the results to phenomena or constructs that were not directly investigated.

4. The hypotheses investigated in the study (last paragraph of the Introduction) seem somewhat trivial given previous results in the literature. With respect to the first hypothesis, while I am not aware of any studies that have specifically tested whether deeper semantic processing of word stimuli results in higher decoding accuracy for semantic categories, it seems to me that this hypothesis must be true at some level. If processing is shallow enough, we expect that the semantic content of the stimuli will be minimally activated, and fMRI activation patterns associated with semantic categories will not be distinguishable from each other. It is useful to

know, however, that the specific manipulation employed in the current study did result in higher decoding accuracies across all ROIs.

It was not clear to me how the second hypothesis (about generalizability) differs from the first. If decoding accuracy is assessed by predicting the category of items not used for training, how are i and ii different? The procedures as described in the Methods do not help clarify the distinction between item categorization and item generalizability. The third hypothesis also seems to logically follow from the first: if concept representations are distributed over different cortical areas (as several studies have indicated), the time course of the activation patterns should be correlated across these areas (because of the time course of the hemodynamic response), and this correlation will, of course, be modulated by the discriminability of the activation patterns. Finally, it is not clear from the Introduction how the fourth hypothesis relates to the main goal of the study regarding the role of depth of processing.

5. No information on MRI acquisition protocols are provided in the Methods section.

6. What were the exact instructions for the mental simulation condition? Were participants instructed to focus on sensory-motor properties (shape, color, size, etc.) or were they encouraged to think of any properties of their own choosing? Were they instructed to engage in mental imagery?

7. How were the 15 ROIs specified? The meta-analysis cited (Binder et al., 2009) lists 7 cortical regions, most of them with bilateral representation, but in the present study all ROIs are in the left hemisphere. Why were right hemisphere regions not included? Also, how were the ROIs defined in the MRI volumes? If an atlas was used, it should be specified in the Methods.

8. "A set of 15 left-lateralized ROIs was pre-specified (see Figure 2) based on a meta-analysis of the semantic system [12] and one anterior temporal lobe (ATL) due to its role as a 'semantic hub'" (p. 61. 50). Although there is considerable evidence for the involvement of the anterior temporal lobe in semantic cognition, its role as a semantic hub has not yet been conclusively established.

9. Encoding model pipeline: "The proportion of variance explained in each voxel was computed for the predictions. An average estimate of the variance explained was calculated. The best possible score is 1. The score can be also negative if the model is worse than random guessing. Voxels that had positive variance explained values were identified for further analysis [42, 45] for each participant, ROI and condition." (p. 11, l. 42). It seems to me that this procedure would bias the results toward models with higher variance (across voxels) of proportion variance explained. Suppose, for instance, that Model A and Model B have the same average score across all voxels in a ROI, with Model A having higher positive scores and lower negative scores (i.e., higher variance of scores). If one looks only at the voxels with positive scores, Model A will erroneously appear to have a higher average score than B.

10. How many catch trials were presented in each block/run?

11. "Out-of-sample Generalization: We then repeated the decoding analyses with a different cross-validation procedure that allowed testing the generalizability of the semantic representations and how this was modulated by the different task contexts. Specifically, the classifier was trained using all the words but leaving a pair of words out from each class. Then the classifier was tested on the left-out pair." (p. 14, l. 10). It is not clear how this analysis evaluates out-of-sample generalization. How is it different from the previous decoding analysis?

12. It is not clear how the results of the informational connectivity analysis should be interpreted. Section 3.4 and Figure 5 describe a complex pattern of increases and decreases in temporal correlations between ROIs, but no attempt is made to explain this pattern or to argue how it is relevant to the goals of the study. On a related note, were any pairs of ROIs significantly more connected in shallow compared to deep processing conditions?

13. Is it possible that the difference in performance between computer vision models and word embedding models reflects differences in the model architectures (e.g., deep learning vs. shallow models) or in their training protocols rather than something about the nature of the neural representations themselves?

14. "Interestingly, the level of informational connectivity [53] observed during mental simulation also indicates that association transmodal cortices interact with multiple regions of the semantic network in terms of the specific information that multivoxel activity patterns carry across time." As I mentioned in point #4 above, it seems to me that the temporal correlations between areas

could be completely explained by the time course of the hemodynamic response, in which case they would not reflect information connectivity.

Author's Response to Decision Letter for (RSOS-192043.R0)

See Appendix A.

RSOS-192043.R1 (Revision)

Review form: Reviewer 1

Is the manuscript scientifically sound in its present form?

Yes

Are the interpretations and conclusions justified by the results?

Yes

Is the language acceptable?

Yes

Do you have any ethical concerns with this paper?

No

Have you any concerns about statistical analyses in this paper?

No

Recommendation?

Accept as is

Comments to the Author(s)

The authors have addressed my comments thoughtfully and effectively. The manuscript is significantly strengthened and will make a valuable contribution to the literature.

Review form: Reviewer 2

Is the manuscript scientifically sound in its present form?

Yes

Are the interpretations and conclusions justified by the results?

No

Is the language acceptable?

Yes

Do you have any ethical concerns with this paper?

No

Have you any concerns about statistical analyses in this paper?

No

Recommendation?

Major revision is needed (please make suggestions in comments)

Comments to the Author(s)

I would like to thank the authors for their careful consideration of the reviewers' comments and for their thoughtful replies. The revised manuscript conveys the goals and methodology of the study much more clearly, and adequately situates it within the existing literature. Although I still have two significant concerns, I think they can be easily addressed. The manuscript would provide a valuable contribution to the cognitive neuroscience of concept representation and language processing.

Specific concerns:

1. In response to my comment regarding the information connectivity analysis, the authors wrote: "It may be argued that the differences in information connectivity may be explained by differences in the temporal correlation between the time courses of the different regions (i.e. the functional connectivity). We argue that if this were the case then we would expect to find the same pattern of differences across conditions at the level of functional connectivity too. To address this concern, we performed functional connectivity analysis as follows. First, the fMRI data was preprocessed and prepared as presented in the Methods sections. Next, a time series was obtained for each ROI by taking the mean of all its voxels across each of the scans. To calculate the functional connectivity between the ROIs, these time-series were correlated using Pearson correlation for each of the pairs of ROIs, and a matrix of functional connectivity was created. This procedure was performed separately for shallow and deep processing sessions resulting in two matrices for each of the participants. Finally, to compare between the functional connectivity in deep and shallow conditions, a paired t-test was conducted with FDR correction. The results are presented in the Supplementary Figure 8 [see below] where it can be seen that there were no pairs of ROIs for which the functional connectivity was found to be significantly different across the deep and shallow conditions. This shows that the pattern of informational connectivity between ROIs cannot be explained by the functional connectivity between them." While I agree with the above, this response does not address the issue to which I was pointing. Univariate functional connectivity is independent of the discriminability of multivoxel activation patterns. However, the temporal correlation in pattern discriminability between two areas is not independent of the discriminability itself. Even if the two areas have the exact same time course of information representation, the lower the discriminability between activation patterns in either area, the more the correlation between their time courses will be dominated by noise, and the lower the correlation will be. Thus, the temporal correlation of word discriminability across different cortical areas depends on (1) how discriminable those patterns are from each other and (2) the time course of the discriminability in each area. If the patterns are less discriminable in the shallow than in the deep processing condition, the inter-areal temporal correlation will necessarily be lower during shallow processing, regardless of "informational connectivity". Since a positive inter-areal temporal correlation in pattern discriminability should be observed as long as the two areas encode similar representations and have similar hemodynamic response functions, it seems to me that what the authors are calling "informational connectivity" is nothing but representational similarity between areas, with the temporal component provided by the time course of the hemodynamic response.

2. In response to my question about why left hemisphere ROIs were not included in the study, the authors stated: "The meta-analysis by Binder et al. (2009) identified 7 cortical regions involved in the processing of words. Binder et al. 2009 further reports evidence from the meta-analysis that the role of these regions in conceptual processing is left-lateralized." The review by Binder et al. (2009) identified areas significantly associated with semantic language processing in both hemispheres, although a larger number of voxels reached significance in the left hemisphere (the red areas in Figure 9 of that paper). That is the only sense in which Binder et al. (2009) claim

the activations are left-lateralized. Several studies published since then (some of which cited in the present manuscript) have confirmed that the right temporoparietal cortex, right precuneus/posterior cingulate, and portions of the right prefrontal cortex are involved in semantic word processing.

I do not think that excluding those right hemisphere areas from the present study would necessarily be a fatal flaw, but the authors should provide a stronger rationale for doing so. As language researchers, we must exercise care to avoid contributing to the widespread misconception that the right hemisphere does not contribute to language semantics.

Decision letter (RSOS-192043.R1)

Dear Professor Soto:

Manuscript ID RSOS-192043.R1 entitled "Decoding and encoding models reveal the role of mental simulation in the brain representation of meaning" which you submitted to Royal Society Open Science, has been reviewed. The comments of the reviewer(s) are included at the bottom of this letter.

Please submit a copy of your revised paper before 10-May-2020. Please note that the revision deadline will expire at 00.00am on this date. If we do not hear from you within this time then it will be assumed that the paper has been withdrawn. In exceptional circumstances, extensions may be possible if agreed with the Editorial Office in advance. We do not allow multiple rounds of revision so we urge you to make every effort to fully address all of the comments at this stage. If deemed necessary by the Editors, your manuscript will be sent back to one or more of the original reviewers for assessment. If the original reviewers are not available we may invite new reviewers.

- Ethics statement

- Data accessibility

- Competing interests

- Authors' contributions

- Acknowledgements

- Funding statement

Kind regards,

Andrew Dunn

on behalf of Dr César Lima (Associate Editor)

Editor comments:

The view of the editors is that your work has substantially improved in this iteration, with only a few comments remaining from the referees - you should pay close attention to these in your

revision: additional opportunities to revise will not be granted unless there are exceptional reasons for doing so. Good luck and thanks for your support of the journal.

Reviewer comments to Author:

Reviewer: 1

Comments to the Author(s)

The authors have addressed my comments thoughtfully and effectively. The manuscript is significantly strengthened and will make a valuable contribution to the literature.

Reviewer: 2

Comments to the Author(s)

I would like to thank the authors for their careful consideration of the reviewers' comments and for their thoughtful replies. The revised manuscript conveys the goals and methodology of the study much more clearly, and adequately situates it within the existing literature. Although I still have two significant concerns, I think they can be easily addressed. The manuscript would provide a valuable contribution to the cognitive neuroscience of concept representation and language processing.

Specific concerns:

1. In response to my comment regarding the information connectivity analysis, the authors wrote: "It may be argued that the differences in information connectivity may be explained by differences in the temporal correlation between the time courses of the different regions (i.e. the functional connectivity). We argue that if this were the case then we would expect to find the same pattern of differences across conditions at the level of functional connectivity too. To address this concern, we performed functional connectivity analysis as follows. First, the fMRI data was preprocessed and prepared as presented in the Methods sections. Next, a time series was obtained for each ROI by taking the mean of all its voxels across each of the scans. To calculate the functional connectivity between the ROIs, these time-series were correlated using Pearson correlation for each of the pairs of ROIs, and a matrix of functional connectivity was created. This procedure was performed separately for shallow and deep processing sessions resulting in two matrices for each of the participants. Finally, to compare between the functional connectivity in deep and shallow conditions, a paired t-test was conducted with FDR correction. The results are presented in the Supplementary Figure 8 [see below] where it can be seen that there were no pairs of ROIs for which the functional connectivity was found to be significantly different across the deep and shallow conditions. This shows that the pattern of informational connectivity between ROIs cannot be explained by the functional connectivity between them." While I agree with the above, this response does not address the issue to which I was pointing. Univariate functional connectivity is independent of the discriminability of multivoxel activation patterns. However, the temporal correlation in pattern discriminability between two areas is not independent of the discriminability itself. Even if the two areas have the exact same time course of information representation, the lower the discriminability between activation patterns in either area, the more the correlation between their time courses will be dominated by noise, and the lower the correlation will be. Thus, the temporal correlation of word discriminability across different cortical areas depends on (1) how discriminable those patterns are from each other and (2) the time course of the discriminability in each area. If the patterns are less discriminable in the shallow than in the deep processing condition, the inter-areal temporal correlation will necessarily be lower during shallow processing, regardless of "informational connectivity". Since a positive inter-areal temporal correlation in pattern discriminability should be observed as long as the two areas encode similar representations and have similar hemodynamic response functions, it seems to me that what the authors are calling "informational connectivity" is nothing but representational similarity between areas, with the temporal component provided by the time course of the hemodynamic response.

2. In response to my question about why left hemisphere ROIs were not included in the study,

the authors stated: "The meta-analysis by Binder et al. (2009) identified 7 cortical regions involved in the processing of words. Binder et al. 2009 further reports evidence from the meta-analysis that the role of these regions in conceptual processing is left-lateralized." The review by Binder et al. (2009) identified areas significantly associated with semantic language processing in both hemispheres, although a larger number of voxels reached significance in the left hemisphere (the red areas in Figure 9 of that paper). That is the only sense in which Binder et al. (2009) claim the activations are left-lateralized. Several studies published since then (some of which cited in the present manuscript) have confirmed that the right temporoparietal cortex, right precuneus/posterior cingulate, and portions of the right prefrontal cortex are involved in semantic word processing.

I do not think that excluding those right hemisphere areas from the present study would necessarily be a fatal flaw, but the authors should provide a stronger rationale for doing so. As language researchers, we must exercise care to avoid contributing to the widespread misconception that the right hemisphere does not contribute to language semantics.

Author's Response to Decision Letter for (RSOS-192043.R1)

See Appendix B.

Decision letter (RSOS-192043.R2)

Dear Professor Soto,

It is a pleasure to accept your manuscript entitled "Decoding and encoding models reveal the role of mental simulation in the brain representation of meaning" in its current form for publication in Royal Society Open Science.

on behalf of Dr César Lima (Associate Editor)
openscience@royalsociety.org

Comments to Author:

Reviewers' Comments to Author:

Reviewer: 1

Comments to the Author(s)

In this work, the authors report an investigation of how conceptual knowledge is represented in the brain. They examine how depth of processing (modulated by task) affects the representation of semantic categories for visual words through an fMRI scan and subsequent analyses. Depth of processing (shallow versus deep) was varied by having participants read words, or mentally simulate the associated features. A series of analyses, including multivariate classification, informational connectivity, and encoding models were employed to compare the conditions. Deep processing increased decoding performance (relative to shallow processing) in regions of the semantic network, and informational connectivity was strengthened between semantic regions. Computer vision models, though not word embeddings, explained variance of the representations across multiple regions.

General comments

This study addresses the interesting question of how neural representations for semantic/conceptual knowledge are affected by current cognitive processing. The manuscript is well written (aside from some minor typos) and the study reports results from an impressive range of methods that are nicely complementary for the study's goals. There are several areas, however, that need to be adjusted before publication. The classification cross-validation procedure should leave out data from untrained runs, rather than timepoints of the same run. Additionally, the informational connectivity comparisons should account for differences in classification performance (see below for one way to do this). Finally, the manuscript would benefit from some further discussion of the difference between processing depth, and the cognitive target of processing. With these changes, the manuscript will make an interesting and useful contribution to the literature.

Specific comments

Major

1) The cross-validation framework holds out items in an 80-20% split (with 300 sets created by randomization). It would be better to hold-out runs as independent testing data. This is because items within the same run can have overlapping hemodynamic responses, so without holding out runs, the training and testing sets are not independent. Similarly, processing such as Z-scoring and exposure to noise from a run can influence a classifier to give above-chance performance. Because of the counterbalanced order of the blocks, it would be best to hold two runs out, to ensure the training set is balanced in this regard (see Snoek et al., 2019, NeuroImage).

Thank you very much for this useful comment. Regarding the issue of the potential overlap of the hemodynamic responses, please note that the inter-trial/stimulus interval was jittered between 10.75 seconds and 12.75 seconds. This long jitter should be sufficient to preclude that the hemodynamic response from one trial does not carry on to the next trial. A note to this effect has been added to the methods. The experiment was designed with this concern in mind and this point has now been explained

more clearly in the paper. However, following your comment, we have run leave-one-run-out cross-validation. Please note that since there are only 4 runs for each of the depth of processing conditions, leaving 2 runs out would reduce quite a lot the training set.

The results of leave-one run out cross-validation are presented as follows. A note to this effect has been added in page 16 (new changes have been tracked in red).

“We also repeated the analyses using a leave-one-run out cross-validation procedure. Here we found significant decoding accuracy in the deep processing condition (see Supplemental Figure 7). However, classification accuracy was at chance in all ROIs in the shallow processing case. Some caution however has to be taken here because there are only 4 cross-validation folds. It is known that leave one out procedures can lead unstable estimates due to across-fold variability, and repeated random splits procedures like the one used above is preferred (Varoquaux et al., 2017).”

Supplementary Figure 7: Summary statistics of decoding accuracy for the different ROIs. The three dotted lines inside each violin are the quartiles. It can be seen that in the shallow processing condition, the decoding of the semantic category (living/non-living) was found to be at chance-level in all ROIs while in the deep condition, it was found to be above-chance and significantly better than the shallow condition in all ROIs.

It can be seen that in the shallow processing condition, the decoding of the semantic category (living/non-living) was found to be at chance-level in all pre-specified ROIs including the FP (50.66±5.46; $t(30) = 0.61$; $p = 0.78$), FFG (50.86±4.76; $t(30) = 0.92$; $p = 0.61$), IPL (50.47±6.38; $t(30) = 0.37$; $p = 0.82$), ITL (51.83±5.25; $t(30) = 1.78$; $p = 0.32$), LOFC (50.32±5.94; $t(30) = 0.28$; $p = 0.83$), MOFC (51.54±6.05; $t(30) = 1.30$; $p = 0.58$), MTL (51.09±4.57; $t(30) = 1.22$; $p = 0.58$), POP (52.25±5.17; $t(30) = 2.22$; $p = 0.26$), POR (51.08±5.37; $t(30) = 1.03$; $p = 0.59$), PTR (50.25±5.91; $t(30) = 0.22$; $p = 0.83$), PHG (49.38±5.55; $t(30) = -0.57$; $p = 0.78$), PCG (49.51±5.65; $t(30) = -0.44$; $p = 0.82$), Precun (52.48±5.37; $t(30) = 2.36$; $p = 0.26$), SFG (52.16±5.65; $t(30) = 1.95$; $p = 0.31$), ATL (48.92±4.92; $t(30) = -1.12$; $p = 0.59$).

On the other hand, in the deep processing condition, the classification of the semantic category was found to be significantly above chance in all pre-specified ROIs (see Figure 1) including FP (52.98±6.43; $t(30) = 2.36$; $p = 0.026$), FFG (66.75±10.47; $t(30) = 8.16$; $p = 1.83e-07$), IPL (66.10±11.09; $t(30) = 7.40$; $p = 2.76e-07$), ITL (64.46±10.87; $t(30) = 6.78$; $p = 8.46e-07$), LOFC (57.66±5.89; $t(30) = 6.64$; $p = 1.04e-06$), MOFC (54.04±5.68; $t(30) = 3.63$; $p = 0.001$), MTL (64.45±9.80; $t(30) = 7.52$; $p = 2.75e-07$), POP (64.30±11.70; $t(30) = 6.23$; $p = 2.55e-06$), POR (55.15±5.43; $t(30) = 4.84$; $p = 7.04e-05$), PTR (61.40±8.57; $t(30) = 6.78$; $p = 8.46e-07$), PHG (58.30±5.58; $t(30) = 7.60$; $p = 2.75e-07$), PCG (58.05±9.78; $t(30) = 4.19$; $p = 0.0004$), Precun (61.87±9.83; $t(30) = 6.16$; $p = 2.72e-06$), SFG (61.82±11.07; $t(30) = 5.44$; $p = 1.58e-05$), and ATL (52.66±5.22; $t(30) = 2.60$; $p = 0.02$).

2) The PCA used to select features (p. 8) should be conducted in the training data of each cross-validation fold (and then applied to the held-out data). This ensures the testing data does not influence the feature selection (and thus final classification performance).

We have clarified this in the methods (see page 10):

“Note that PCA was performed on the training set; then the trained PCA was used to extract components in the test data and its classification performance was assessed. This procedure was repeated separately for each of the 300 sets, and the mean of corresponding accuracies was collected for each of the ROIs and participants.”

3) How might the focus of cognitive processing influence the differences observed between deep and shallow conditions. For example, the process target of the shallow task is reading the words, versus imaging properties of the item. It seems that two differences are present in the conditions – between the depth of processing, and between the particular characteristics emphasized when encountering the words. This is somewhat unavoidable, but it would be worth discussing what this means for the conclusions about how depth processing modulates the representation of meaning, rather than reflecting differences in focus of attention, irrespective of ‘depth’ per se. For instance, a third equally deep condition (focusing on other characteristics) might in theory help with this.

We thank the reviewer for the thoughtful remark. To address this issue, we have added a note to the Discussion (see page 25) to elaborate on the limitations of the experimental manipulations.

“One limitation of the current study is that task manipulation was relatively coarse grained, i.e. mental simulation vs. covert reading of the items. Our goal here was to examine how the depth of information processing influenced the brain representation of meaning and, accordingly, we elected to compare task contexts that differed in the need to consciously represent the meaning of the items. This is important because there has been little research on the role of task factors at influencing the brain representation of meaning. Seminal studies on neural correlates of the levels of processing effect (Kapur et al., 1994) used univariate approaches that are not well suited to understand the brain representation of meaning.

The effect of mental simulation on classification accuracy is however unlikely to be due to differences in the degree of attentional allocation to the items and differences in task engagement. First, catch trials were present across the different task contexts and performance was similar, suggesting that participants deployed attention and were engaged with the tasks to a similar extent. Instead, we interpret the effects of mental simulation in terms of the re-enactment of the sensorimotor processes associated with the perception of the referents of the words. Nevertheless, we acknowledge that further studies tapping on more fine-grained manipulations of semantic processing (i.e. lexical vs. concreteness decisions or feature vs category verification tasks) and sensory experience (i.e. visual vs auditory) may be useful to pinpoint more precisely how the nature of the specific task shapes how the brain encodes meaning.”

4) The comparison of informational connectivity strengths between the two conditions is very interesting, but it is necessary to control for the significantly different levels of classification that are reported elsewhere. When one condition has superior classification (here, deep processing), it is more likely by chance that two regions will be more correlated (due to having more higher values in common). One way is to calculate statistical significance by permuting (scrambling) one region’s values to create a null distribution of informational connectivity. This will ensure that the overall decoding performance (or distance from the classifier’s bounds) is not inflating the deep-processing’s connectivity matrix, relative to the shallow-processing for the same basis of differences in overall performance.

We thank the reviewer for this very useful comment. We have included new analyses and a new paragraph in the Results (see page 20) to respond to this point: *“It could be argued that the overall high decoding performance in the deep condition may be inflating the corresponding informational connectivity score matrices. We conducted the following analysis to address this. For each pair of ROIs, we defined the ‘ground truth’ value of Pearson’s correlation, namely, the correlation between the unshuffled MVP discriminability vectors of two given ROIs. Next, we randomly shuffled the first ROI’s discriminability vector 10,000 times, and each time we correlated the shuffled vector to the second ROI’s unshuffled vector, and thus created a null distribution of correlation values. Next, we computed the difference between the ground-truth and the mean of the null distribution and divided it by the variance of the distribution, thus calculating a measure of effect size for each pair of ROIs. Note this is independent of the level of classification performance. To compare this effect size between deep and shallow conditions, paired t-tests with FDR correction were run across subjects. We found that for all pairs of ROIs, this effect size was statistically significantly greater in the deep compared to the shallow condition, except for the pair involving the precuneus and superior frontal gyrus ($p = 0.053$). This shows that the*

higher level of decoding accuracy in the deep processing condition can not account for the informational connectivity results.”

5) Please list the scanning parameters. For example, orientation of slice acquisition, TR, voxel size, etc.

We apologise for missing this important information. The details below have now been added to the paper in the MRI acquisition section (see page 7).

“A SIEMENS’s Magnetom Prisma-fit scanner, with 3 Tesla magnet and 64-channel head coil, was used to collect, for each participant, one high-resolution T1-weighted structural image and eight functional images (corresponding to eight runs/sessions). In each fMRI session, a multiband gradient-echo echo-planar imaging sequence with multi-band acceleration factor of 6, resolution of 2.4 x 2.4 x 2.4mm³, TR of 850 ms, TE of 35 ms and bandwidth of 2582 Hz/Px was used to obtain 520 3D volumes of the whole brain (66 slices; FOV = 210mm). The visual stimuli were projected on an MRI-compatible out-of-bore screen using a projector placed in the room adjacent to the MRI-room. A small mirror, mounted on the head coil, reflected the screen for presentation to the participants. The head coil was also equipped with a microphone that enabled the participants to communicate with the experimenters in between the sessions.”

Minor

6) The data appears to be behind a request form.

We have uploaded the data to Dryad (<https://datadryad.org/>) following instructions from the journal. We understand this is the normal procedure but we would be happy for the data not to be behind a request form.

7) It would be helpful to translate the words into English (page 5). This is completed in the figure, but not the list of words

This has been done (see page 7). Apologies for not including it earlier.

“Corresponding English translations were the following. Living words: tiger, rooster, dog, sheep, pig, gorilla, donkey, mare, squirrel, rabbit, hen, horse, whale, turtle, panther, camel, elephant, kangaroo. Non-living words: wrench, pencil, scissors, needle, clamp, saw, nail, brush, pliers, nut, knife, brush, drill, blowtorch, screw, spoon, hammer, knife.”

8) Were all the stimuli known to participants? It would be worth reporting familiarity statistics from an available norming dataset, or an independent set of participants. This is important because it is naturally more difficult to simulate features for concepts that are not as familiar.

We thank the reviewer for this comment. We have addressed this in page 7, Methods section. We gathered familiarity ratings for the actual Spanish stimuli using EsPal Database (Duchon et al. 2013). This database includes familiarity ratings between 1 and 7. There were a few words for which the rating was not found. However, if we consider only the words for which it was found, familiarity ratings were very similar between categories (mean living: 5.94; mean non-living: 5.98). We also looked at the subjective

familiarity ratings of the English translations of the same words. These were obtained using MRC Psycholinguistic Database (Coltheart 1981). This database includes ratings between 100 and 700. There were again a few words for which the rating was not found, however, we again observed that familiarity ratings are very similar between categories (mean living: 511; mean non-living: 498). A note to this effect has been added to the Methods.

References:

M. Coltheart (1981), The MRC Psycholinguistic Database, *Quarterly Journal of Experimental Psychology*, 33A, 497-505.

Duchon, A., Perea, M., Sebastián-Gallés, N., Martí, A., & Carreiras, M. (2013). EsPal: One-stop shopping for Spanish word properties. *Behavior research methods*, 45(4), 1246-1258.

9) On page 20, the authors state that “The depth of processing associated with mental simulation therefore triggered a general broadcasting effect of the specific content being represented”. What does “general broadcasting” mean here? Different terminology might make the implications clearer to the reader. Do you mean the signal is sent elsewhere? If so, why?

Thanks for the remark. Indeed, that is what we meant and we have tried to clarify this issue in the new paragraph (see page 24)..

“We propose that the depth of information processing associated with mental simulation involved the broadcasting of the information across a distributed set of areas of the semantic network, hence making information globally available and consciously accessible. The global availability of information across the brain networks involved in semantic processing may be critical for the successful retrieval and manipulation of conceptual knowledge to guide thought and behaviour.”

Reviewer: 2

This study investigates how “depth of processing” affects the neural representation of semantic categories as measured through fMRI multi-voxel pattern analysis. Participants performed a visual word recognition task with two conditions: in the “shallow processing” condition participants were instructed to passively read the words, while in the “deep processing” condition they were instructed to “mentally simulate the properties associated with the word”. Conditions were blocked by scanning runs. The stimuli consisted of 36 concrete nouns (18 animal names and 18 tool names). Participants provided overt responses only to catch trials, in which a number word was presented. Three different analyses were performed: (1) A support vector machine pattern classification analysis was used to assess the extent to which semantic category information (living/nonliving) was present in the fMRI activation patterns in each condition (“shallow” vs. “deep” processing). This analysis was performed separately in 15 pre-defined regions of interest (ROIs). It revealed that, in 13 of the 15 ROIs, classification accuracy was higher in the deep processing condition. (2) An informational connectivity analysis was used to characterize the temporal correlation of semantic

category discriminability between ROIs over the course of the trial. The purpose of this analysis was to investigate whether this informational connectivity between ROIs varies across deep and shallow processing conditions. It revealed that the temporal correlation between regions was significantly higher in the deep processing condition for several pairs of ROIs. The authors do not report whether any pairs of ROIs were significantly more connected in shallow compared to deep processing conditions. (3) Finally, an encoding model analysis compared the proportion of variance explained by distributional word embedding models (Fast Text, GloVe, and Word2vec) and computer vision models (VGG19, MobileNetV2, and DenseNet169), separately for each ROI and each condition (shallow vs. deep processing). No differences were found between the three word embedding models or between the three computer vision models, but computer vision models performed significantly better than word embedding models. The effect of condition, however, was mixed: “the advantage of computer vision models over word embeddings models was higher in the deep processing condition relative to the shallow processing in PCG, PHG, and POP, while the opposite pattern was observed in FFG, IPL, and ITL.”

The study addresses an important issue in the cognitive neuroscience of conceptual thinking and language processing, namely, the nature of the code underlying the neural representation of word meaning. However, I do have substantial concerns about its specific goals, methods, and interpretation of results, which are listed below.

We thank the reviewer for the feedback.

Major concerns:

1. The manuscript does not appropriately contextualize the study within the neuroimaging literature on concept representation (e.g., Devereux, B. J. et al., 2013. Representational similarity analysis reveals commonalities and differences in the semantic processing of words and objects. *The Journal of Neuroscience*, 33(48); Anderson, A. J. et al., 2015. Reading visually embodied meaning from the brain: Visually grounded computational models decode visual-object mental imagery induced by written text. *NeuroImage*, 120; Fernandino, L. et al., 2016. Heteromodal Cortical Areas Encode Sensory-Motor Features of Word Meaning. *Journal of Neuroscience*, 36(38); Martin, C. B. et al., 2018. Integrative and distinctive coding of visual and conceptual object features in the ventral visual stream. *eLife*, 7, 1493; Carota, F. et al., 2017. Representational Similarity Mapping of Distributional Semantics in Left Inferior Frontal, Middle Temporal, and Motor Cortex. *Cerebral Cortex*, 27(1); Devereux, B.J. et al., 2018. Integrated deep visual and semantic attractor neural networks predict fMRI pattern-information along the ventral object processing pathway. *Sci Rep* 8, 10636; Anderson, A. J. et al. (2019). An Integrated Neural Decoder of Linguistic and Experiential Meaning. *The Journal of Neuroscience*, 39(45)), levels of processing (e.g., Nyberg, L., 2002. Levels of processing: A view from functional brain imaging. *Memory*, 10(5)) or mental imagery (e.g., Daselaar, S. M. et al., 2010. Modality-specific and modality-independent components of the human imagery system. *NeuroImage*, 52(2); Huijbers, W. et al., 2011. Imagery and retrieval of auditory and visual information: neural correlates of successful and unsuccessful performance. *Neuropsychologia*, 49(7)). The Introduction only cites a few hand-picked studies that, in my opinion, are not as relevant to the current study as some of the references above.

We thank the reviewer for bringing this important literature to our attention. This has been useful to re-work the introduction and contextualise the study better. Please see the new changes highlighted in red in the revised introduction section (see pages 3-5).

2. There is a discrepancy between the wide scope of the study's claims, as stated in the title, abstract, introduction, and discussion, and the narrow scope of the semantic manipulations that were actually employed. In particular, the assessment of semantic discriminability was restricted to the distinction between two categories (animals and tools). It is not clear that the effects of task demands found for this particular categorical distinction can be generalized to other categorical distinctions, much less to "representational spaces of conceptual knowledge" (p. 3, l. 37) in general. I suggest the authors rephrase their claims to reflect the actual scope of the investigation.

We agree that this is a limitation of the current study. We have toned down claims regarding "brain representational spaces of conceptual knowledge" and we have also added a note in the Discussion (see page 25) acknowledging the limitations of the present work in this regard.

"Nevertheless, we acknowledge that further studies tapping on more fine-grained manipulations of semantic processing (i.e. lexical vs. concreteness decisions or feature vs category verification tasks) and sensory experience (i.e. visual vs auditory) including a wider range of stimulus classes, may be useful to pinpoint more precisely how the nature of the specific task shapes how the brain encodes meaning. Additional work needs to be conducted to make further determinations."

3. While the study's main goal is to investigate "the role of depth of processing in the brain representation of meaning", the term "depth of processing" is never defined. In the Introduction, the authors cite three studies as examples of deep processing, all from the same group and using the same task: participants are presented with a written word denoting a concrete concept and, simultaneously, a line drawing of the same concept, and are instructed to think about salient properties of the concept. As examples of shallow processing, they cite a study by Simanova et al. (2012), in which subjects were presented with four types of stimuli (visual words, auditory words, colored photographs, and non-verbal natural sounds) and instructed to "judge whether each stimulus within a block was semantically consistent with the others", and a study by Huth et al. (2016), in which participants listened to pre-recorded narrative stories. These examples do not provide a clear picture of what the authors mean by "depth of processing". It is, therefore, difficult to evaluate their claim that "previous research has not determined the role of the depth of processing on the brain representation of concepts" (p. 3, l. 39). In the present study, "depth of processing" is operationalized as the difference between (a) passively reading an isolated concrete noun and (b) engaging in "mental simulation" of the noun. Again, it is not clear what "mental simulation" refers to, since the actual instructions to participants are not reported in the manuscript, but if we assume that in this condition participants were instructed to focus of the sensory-motor properties of the concept named, then mental imagery is the process that is actually under investigation.

We fully agree. We have clarified what we meant by depth of processing in the context of the present study; please see page 4 where we note.

“In this study, the depth of processing was operationalised as the difference between covertly reading a word (shallow processing) and mentally simulating the properties of the concept (deep processing). Mental simulation here refers to the ability to imagine or re-enact modality specific representations. Although this manipulation differs from the seminal experimental framework of ‘levels of processing’ (cf. Craik and Tulving, 1972), which used more targeted tasks tapping on semantic vs. phonemic/orthographic properties judgements, our experimental manipulation is nevertheless consistent with varying depths of processing in that mental simulation requires deeper access to the meaning of the word items, while the condition of shallow processing focuses mainly on phonological representations.

Further, we have elected to change the title to *“Decoding and encoding models reveal the role of mental simulation in the brain representation of meaning”*.

It is far from clear, however, that the effects of this mental imagery manipulation on the semantic discriminability of the stimulus categories can be generalized to all processes (or manipulations) that could reasonably fall under the umbrella of “depth of processing”. The cognitive neuroscience literature on word processing contains a wide variety of semantic manipulations that can be construed as different levels of processing depth, such as phonological decision, lexical decision, familiarity rating, concreteness rating, feature verification, category membership judgment, synonym matching, etc. I suggest the authors provide a clearer definition of the phenomenon under investigation and exercise more restraint in extrapolating the implications of the results to phenomena or constructs that were not directly investigated.

Thanks for the thoughtful remark. We have included additional discussion in page 25 to address this point.

“One limitation of the current study is that task manipulation was relatively coarse grained, i.e. mental simulation vs covert reading of the items. Our goal here was to examine how the depth of information processing influenced the brain representation of meaning and, accordingly, we elected to compare task contexts that differed in the need to consciously represent the meaning of the items. This is important because there has been little research on the role of task factors at influencing the brain representation of meaning. Seminal studies on neural correlates of the levels of processing effect (Kapur et al., 1994) used univariate approaches which are not well suited to understand the brain representation of meaning.

The effect of mental simulation on classification accuracy is however unlikely to be due to differences in the degree of attentional allocation to the items and differences in task engagement. First, catch trials were present across the different task contexts and performance was similar, suggesting that participants deployed attention and were engaged with the tasks to a similar extent. Instead, we interpret the effects of mental simulation in terms of the re-enactment of the sensorimotor processes associated with the perception of the referents of the words. Nevertheless, we acknowledge that further studies tapping on

more fine-grained manipulations of semantic processing (i.e. lexical vs. concreteness decisions or feature vs category verification tasks) and sensory experience (i.e. visual vs auditory) may be useful to pinpoint more precisely how the nature of the specific task shapes how the brain encodes meaning”.

We have also clarified the specific instructions given to participants in page 6:

“During these 4 s delay, depending on the condition, the participants were asked to either covertly read the word (shallow processing condition) or mentally simulate the properties associated with the word, henceforth, the deep processing condition. Specifically, in the shallow processing blocks, we asked participants to read and repeat the word in their mind (i.e. in the phonological loop) until the red asterisk is displayed. This was done in an attempt to equate the level of task load across task conditions. In the deep processing, participants were required to engage in mental simulation of the words, namely, they were asked to re-represent the sensory-motor experiences (i.e item shape, color, auditory, action- and context- related features etc) associated with the word during the delay period until the red asterisk is displayed. They were instructed to use similar mental simulations in subsequent presentations of the same word“.

4. The hypotheses investigated in the study (last paragraph of the Introduction) seem somewhat trivial given previous results in the literature. With respect to the first hypothesis, while I am not aware of any studies that have specifically tested whether deeper semantic processing of word stimuli results in higher decoding accuracy for semantic categories, it seems to me that this hypothesis must be true at some level. If processing is shallow enough, we expect that the semantic content of the stimuli will be minimally activated, and fMRI activation patterns associated with semantic categories will not be distinguishable from each other. It is useful to know, however, that the specific manipulation employed in the current study did result in higher decoding accuracies across all ROIs.

Given that the role of task factors on the brain representation remains to be determined, we disagree that the hypotheses seem trivial. We have however tried to improve their theoretical justification and conceptualise the hypotheses better in the context of the literature.

We agree with the reviewer that, in principle, semantic decoding accuracy is expected to be higher during deep compared to shallow processing. However, as the referee notes, it remains unclear whether the depth of processing associated with mental simulation has a local or a generalised, distributed effect on the level of decoding accuracy across the regions of the semantic network. We have added a note to this effect in the introduction.

It was not clear to me how the second hypothesis (about generalizability) differs from the first. If decoding accuracy is assessed by predicting the category of items not used for training, how are i and ii different? The procedures as described in the Methods do not help clarify the distinction between item categorization and item generalizability.

We have now made it clearer both in the Methods (page 10) and Results (page 15) that the classification analyses used two types of cross-validation. In the first type, independent scans were used for training and testing (i.e., training occurred with partitions containing 80% of the data and testing with the remaining 20%), the scans could both refer to the same words. In the second type of cross-validation, the training and testing partitions did not refer to similar words (i.e. scans associated with a pair of words -- living and non-living -- was left out for testing). Hence, this second type of cross-validation involved out-of-sample generalization). Interestingly, we found that during shallow processing significant classification accuracy occurred in the former but not in the latter case, indicating that the brain representations of meaning are not generalizable across concepts of the same class during shallow processing. Out-of-sample generalization was robust when the depth of processing was higher, during mental simulation.

The third hypothesis also seems to logically follow from the first: if concept representations are distributed over different cortical areas (as several studies have indicated), the time course of the activation patterns should be correlated across these areas (because of the time course of the hemodynamic response), and this correlation will, of course, be modulated by the discriminability of the activation patterns.

Please note that the concept representations were distributed in *both* shallow and deep processing conditions, when the first type of cross-validation was used as indicated above. However, we have now acknowledged in the introduction that any confounding effects due to the potential correlation of the time courses across ROIs will be addressed (see page 5). Accordingly, we conducted functional connectivity analyses among the ROIs and now report in page 19 that there are no significant differences in functional connectivity across the shallow and deep processing conditions.

“It may be argued that the differences in information connectivity may be explained by differences in the temporal correlation between the time courses of the different regions (i.e. the functional connectivity). We argue that if this were the case then we would expect to find the same pattern of differences across conditions at the level of functional connectivity too. To address this concern, we performed functional connectivity analysis as follows. First, the fMRI data was preprocessed and prepared as presented in the Methods sections. Next, a time series was obtained for each ROI by taking the mean of all its voxels across each of the scans. To calculate the functional connectivity between the ROIs, these time-series were correlated using Pearson correlation for each of the pairs of ROIs, and a matrix of functional connectivity was created. This procedure was performed separately for shallow and deep processing sessions resulting in two matrices for each of the participants. Finally, to compare between the functional connectivity in deep and shallow conditions, a paired t-test was conducted with FDR correction. The results are presented in the Supplementary Figure 8 [see below] where it can be seen that there were no pairs of ROIs for which the functional connectivity was found to be significantly different across the deep and shallow conditions. This shows that the pattern of informational connectivity between ROIs cannot be explained by the functional connectivity between them.”

Finally, it is not clear from the Introduction how the fourth hypothesis relates to the main goal of the study regarding the role of depth of processing.

We have improved the rationale in the following paragraph in the Introduction: *“Regarding the encoding models, we tested different encoding models based on word embeddings and computer vision models fitted with the image referents of the words to examine the properties of the corresponding brain representations, in particular, how semantic/syntactic vs. visual properties associated with the words are encoded across the shallow and deep processing conditions. Critically, our experimental paradigm allowed us to test the role of task factors in encoding-related activity which, as noted above, prior studies did not address. We reasoned that if simulation processes for word concepts occur automatically in perceptual systems (Barsalou et al. 2008) then we would expect that computer vision models explain brain responses during semantic processing to a similar extent across the different depth of processing conditions. We further determined the extent to which different types of features (i.e. semantic/syntactic vs. perceptual, i.e. visual) are encoded in brain activity. According to perceptual symbols theory, conceptual knowledge is supported by simulation processes in perceptual systems and this simulation is thought to occur automatically without the need of committing the information into a conscious working memory system (Barsalou et al. 2008).*

5. No information on MRI acquisition protocols are provided in the Methods section.

We thank the reviewer for this comment. This information has now been added to the paper in pages 7.

“A SIEMENS’s Magnetom Prisma-fit scanner, with 3 Tesla magnet and 64-channel head coil, was used to collect, for each participant, one high-resolution T1-weighted structural image and eight functional images (corresponding to eight runs/sessions). In each fMRI session, a multiband gradient-echo echo-planar imaging sequence with multi-band acceleration factor of 6, resolution of $2.4 \times 2.4 \times 2.4\text{mm}^3$, TR of 850 ms, TE of 35 ms and bandwidth of 2582 Hz/Px was used to obtain 520 3D volumes of the whole brain (66 slices; FOV = 210mm). The visual stimuli were projected on an MRI-compatible out-of-bore

screen using a projector placed in the room adjacent to the MRI-room. A small mirror, mounted on the head coil, reflected the screen for presentation to the participants. The head coil was also equipped with a microphone that enabled the participants to communicate with the experimenters in between the sessions.”

6. What were the exact instructions for the mental simulation condition? Were participants instructed to focus on sensory-motor properties (shape, color, size, etc.) or were they encouraged to think of any properties of their own choosing? Were they instructed to engage in mental imagery?

Please see our response to point 3 above, last paragraph.

7. How were the 15 ROIs specified? The meta-analysis cited (Binder et al., 2009) lists 7 cortical regions, most of them with bilateral representation, but in the present study all ROIs are in the left hemisphere. Why were right hemisphere regions not included? Also, how were the ROIs defined in the MRI volumes? If an atlas was used, it should be specified in the Methods.

We thank the reviewer for this comment. The meta-analysis by Binder et al. (2009) identified 7 cortical regions involved in the processing of words. Binder et al. 2009 further reports evidence from the meta-analysis that the role of these regions in conceptual processing is left-lateralized. However, the size of these 7 regions was quite large. Accordingly, we elected to derive 15 more fine-grained ROIs contained within the 7 cortical identified by Binder (2009), in order to gain more anatomical resolution and at the same time mitigate the curse of dimensionality issue in the multivariate classification analyses. Further, the Regions of interest Methods section in page 8 notes that an automatic segmentation of the high-resolution structural scan was done with FreeSurfer and the resulting individual anatomical masks were transformed to functional space using 7 DoF linear registrations implemented in FSL FLIRT and binarized prior to further analyses, which were performed in native BOLD space.

8. “A set of 15 left-lateralized ROIs was pre-specified (see Figure 2) based on a meta-analysis of the semantic system [12] and one anterior temporal lobe (ATL) due to its role as a ‘semantic hub’” (p. 6 l. 50). Although there is considerable evidence for the involvement of the anterior temporal lobe in semantic cognition, its role as a semantic hub has not yet been conclusively established.

Thank you for the note. We now simply refer to the role of the ATL in semantic cognition.

9. Encoding model pipeline: “The proportion of variance explained in each voxel was computed for the predictions. An average estimate of the variance explained was calculated. The best possible score is 1. The score can be also negative if the model is worse than random guessing. Voxels that had positive variance explained values were identified for further analysis [42, 45] for each participant, ROI and condition.” (p. 11, l. 42). It seems to me that this procedure would bias the results toward models with higher variance (across voxels) of proportion variance explained. Suppose, for instance, that Model A and Model B have the same average score across all voxels in a ROI, with Model A having higher positive scores and lower negative scores (i.e., higher variance of

scores). If one looks only at the voxels with positive scores, Model A will erroneously appear to have a higher average score than B.

Thank you very much for the comment on the encoding model pipeline. We acknowledge this is a potential limitation of this approach, yet we have followed the same modeling pipeline established by prior studies (i.e. Miyawaki et al., 2008, Holdgraf et al., 2017, Güçlü and van Gerven, 2017; Naselaris et al., 2011). There are two conventional ways to analyze encoding model results based on the goodness of fit and voxel-wise brain mapping. Due to a large number of ROIs, we decided to use the goodness of fit to summarize the encoding model results so that we could compare the model performance across conditions and ROIs. We are aware of the potential bias towards models with higher variance explained in the comment above. However, we now present additional Supplementary Results referred to in the last paragraph of page 20 in the manuscript and Supplementary Figures 13, 14 and 15. These results show that computer vision models generally explain the variance of *more voxels* than the word embedding model. The Figure below shows the number of positive variance-explained voxels across vision and word embedding models for a given ROI and condition. ROIs are color-coded and conditions are coded in different markers (i.e. X: deep processing; O: shallow processing).

Next, we show in the Figure below the statistically significant differences in the number of positive variance explained voxels between computer vision models and word embedding models for each ROI and condition (FDR-corrected for multiple comparisons). *: $p < 0.05$, **: $p < 0.01$, ***: $p < 0.0001$.

Additionally, as shown in the last Figure below, if a given voxel was positively explained by the word embedding models, it is more likely that the very same voxel was better explained by the computer vision models. ROIs are color-coded and conditions are coded in different markers (i.e. X: deep processing; O: shallow processing). The Figure illustrates the variance explained of individual voxels for all ROIs and conditions. Voxels that cannot be positively explained by either the computer vision nor the word embedding models are shown by black circles. A few (~100 voxels for all subjects, ROIs, and conditions) that have extreme negative variance explained (< -1000) are not shown on the figure.

To quantify this observation, we modeled the probability of computer vision models explaining more variance than word embedding models of a given single voxel using a binomial distribution. This is depicted in the Figure below as θ . The prior probability of the above was given by a prior distribution centered at 0.5. For a given ROI and condition, voxels that were better explained by computer vision models were labeled “1” and “0” otherwise. The posterior probability was computed by multiplying the prior probability and the likelihood of “1”. As shown in the Figure below, for a given voxel in almost all the ROIs and conditions, it was indeed more likely that computer vision models explained more variance than word embedding models, the only exception being the frontal pole in the shallow processing condition.

Finally, we note that in doing these revisions we realised that the graphical illustration of the advantage of computer vision models over the word embedding models during deep relative to shallow processing (e.g. Figure 8) was inaccurate and hence we have amended this. It turns out that the advantage of computer vision models over word embedding models was higher during mental simulation relative to the shallow condition in all cases in which there were significant differences between conditions. In particular, during deep processing computer vision models explained more variance in areas of the ventral visual pathway, including the fusiform, inferior temporal and inferior parietal cortex, and also in the posterior cingulate and inferior frontal cortex. A note to this effect was added to the Discussion and the Figures have been also revised.

References

- Miyawaki, Y., Uchida, H., Yamashita, O., Sato, M. A., Morito, Y., Tanabe, H. C., ... & Kamitani, Y. (2008). Visual image reconstruction from human brain activity using a combination of multiscale local image decoders. *Neuron*, 60(5), 915-929.
- Holdgraf, C. R., Rieger, J. W., Micheli, C., Martin, S., Knight, R. T., & Theunissen, F. E. (2017). Encoding and decoding models in cognitive electrophysiology. *Frontiers in systems neuroscience*, 11, 61.
- Güçlü, U., & van Gerven, M. A. (2017). Modeling the dynamics of human brain activity with recurrent neural networks. *Frontiers in computational neuroscience*, 11, 7.
- Naselaris, T., Kay, K. N., Nishimoto, S., & Gallant, J. L. (2011). Encoding and decoding in fMRI. *Neuroimage*, 56(2), 400-410.

10. How many catch trials were presented in each block/run?

A minimum of zero and a maximum of two catch trials were presented in each run; however, it was made sure that the total number of catch trials presented in shallow processing condition (mean = 4.05 ± 0.22) are same as that in the deep condition (mean = 3.85 ± 0.59) with no statistically significant difference ($p = 0.104$). A note to this effect has been added in page 6.

11. “Out-of-sample Generalization: We then repeated the decoding analyses with a different cross-validation procedure that allowed testing the generalizability of the semantic representations and how this was modulated by the different task contexts. Specifically, the classifier was trained using all the words but leaving a pair of words out from each class. Then the classifier was tested on the left-out pair.” (p. 14, l. 10). It is not clear how this analysis evaluates out-of-sample generalization. How is it different from the previous decoding analysis?

This has been clarified above in point 4.

12. It is not clear how the results of the informational connectivity analysis should be interpreted. Section 3.4 and Figure 5 describe a complex pattern of increases and decreases in temporal correlations between ROIs, but no attempt is made to explain this pattern or to argue how it is relevant to the goals of the study. On a related note, were any pairs of ROIs significantly more connected in shallow compared to deep processing conditions?

The figure below shows the difference of the connectivity matrices, and corresponding statistical significance plot. It can be seen that there was only one pair of ROIs in which the level of informational connectivity was higher in the shallow compared to deep processing condition i.e. frontal pole and middle temporal lobe. This is now noted in the Results. Clearly, the pattern shows that the level of informational connectivity was enhanced in the deep processing condition.

We have also added a new paragraph regarding the interpretation of the informational connectivity analysis.

“We propose that the depth of information processing associated with mental simulation involved the broadcasting of the information across a distributed set of areas of the semantic network, hence making information globally available and consciously accessible. The global availability of information across the brain networks involved in semantic processing may be critical for the successful retrieval and manipulation of conceptual knowledge to guide thought and behaviour.”

13. Is it possible that the difference in performance between computer vision models and word embedding models reflects differences in the model architectures (e.g., deep learning vs. shallow models) or in their training protocols rather than something about the nature of the neural representations themselves?

Thank you for the comment. We were aware of this potential confound and we mitigate it in the following way. Please note that we matched the dimensionality of the representations from both word embedding and computer vision when we extracted features for encoding. As noted in the paper, because the word embedding models represent individual words by 300-element vectors while the computer vision models using much higher dimensions, we fine-tuned the computer vision models using their pre-trained convolutional layers stacked with two new fully-connected dense layers. One of the dense layers was a 300-unit dense layer re-ensembling features extracted from the convolutional layers, and the other was a 96-unit dense layer predicting classes of the data used for fine-tuning. We used the 96 categories of Caltech101 data to fine-tune the newly added layers while the convolutional layers were unchanged. Additionally, there was no significant difference in encoding performance by each of their extracted features (Figure 6 of the paper), even though there were significant different architectures of depth between the computer vision models. Thus, the feature representations of the computer vision models were also 300-element vectors and accordingly it is unlikely that the difference in performance between models reflects differences in the depth of the model.

14. “Interestingly, the level of informational connectivity [53] observed during mental simulation also indicates that association transmodal cortices interact with multiple regions of the semantic network in terms of the specific information that multivoxel activity patterns carry across time.” As I mentioned in point #4 above, it seems to me that the temporal correlations between areas could be completely explained by the time course of the hemodynamic response, in which case they would not reflect information connectivity.

As noted above in response to point 4, we conducted functional connectivity analyses among the ROIs showing no significant differences in functional connectivity across the shallow and deep processing conditions. This indicates that the pattern of informational connectivity between ROIs cannot be merely explained by the levels of temporal correlation, i.e. functional connectivity, between the ROIs. Please see Supplementary Figure 8.

Appendix B

Reviewer #2 Comments to the Author(s)

I would like to thank the authors for their careful consideration of the reviewers' comments and for their thoughtful replies. The revised manuscript conveys the goals and methodology of the study much more clearly, and adequately situates it within the existing literature. Although I still have two significant concerns, I think they can be easily addressed. The manuscript would provide a valuable contribution to the cognitive neuroscience of concept representation and language processing.

Specific concerns:

1. In response to my comment regarding the information connectivity analysis, the authors wrote: "It may be argued that the differences in information connectivity may be explained by differences in the temporal correlation between the time courses of the different regions (i.e. the functional connectivity). We argue that if this were the case then we would expect to find the same pattern of differences across conditions at the level of functional connectivity too. To address this concern, we performed functional connectivity analysis as follows. First, the fMRI data was preprocessed and prepared as presented in the Methods sections. Next, a time series was obtained for each ROI by taking the mean of all its voxels across each of the scans. To calculate the functional connectivity between the ROIs, these time-series were correlated using Pearson correlation for each of the pairs of ROIs, and a matrix of functional connectivity was created. This procedure was performed separately for shallow and deep processing sessions resulting in two matrices for each of the participants. Finally, to compare between the functional connectivity in deep and shallow conditions, a paired t-test was conducted with FDR correction. The results are presented in the Supplementary Figure 8 [see below] where it can be seen that there were no pairs of ROIs for which the functional connectivity was found to be significantly different across the deep and shallow conditions. This shows that the pattern of informational connectivity between ROIs cannot be explained by the functional connectivity between them."

While I agree with the above, this response does not address the issue to which I was pointing. Univariate functional connectivity is independent of the discriminability of multivoxel activation patterns. However, the temporal correlation in pattern discriminability between two areas is not independent of the discriminability itself. Even if the two areas have the exact same time course of information representation, the lower the discriminability between activation patterns in either area, the more the correlation between their time courses will be dominated by noise, and the lower the correlation will be. Thus, the temporal correlation of word discriminability across different cortical areas depends on (1) how discriminable those patterns are from each other and (2) the time course of the discriminability in each area. If the patterns are less discriminable in the shallow than in the deep processing condition, the inter-areal temporal correlation will necessarily be lower during shallow processing, regardless of "informational connectivity". Since a positive inter-areal temporal correlation in pattern discriminability should be observed as long as the two areas encode similar representations and have similar hemodynamic response functions, it seems to me that what the authors are calling "informational connectivity" is nothing but representational similarity between areas, with the temporal component provided by the time course of the hemodynamic response.

We believe this issue was already addressed the prior round of revisions in response to point 4 of reviewer #1 in the previous rounds of reviews, where the reviewer noted:

(4) The comparison of informational connectivity strengths between the two conditions is very interesting, but it is necessary to control for the significantly different levels of classification that are reported elsewhere. When one condition has superior classification (here, deep processing), it is more likely by chance that two regions will be more correlated (due to having more higher values in common). One way is to calculate statistical significance by permuting (scrambling) one region's values to create a null distribution of informational connectivity. This will ensure that the overall decoding performance (or distance from the classifier's bounds) is not inflating the deep-processing's connectivity matrix, relative to the shallow-processing for the same basis of differences in overall performance.

To respond to this query we included new analyses and a new paragraph in the Results (see page 20) to respond to this point: “It could be argued that the overall high decoding performance in the deep condition may be inflating the corresponding informational connectivity score matrices. We conducted the following analysis to address this. For each pair of ROIs, we defined the ‘ground truth’ value of Pearson’s correlation, namely, the correlation between the unshuffled MVP discriminability vectors of two given ROIs. Next, we randomly shuffled the first ROI’s discriminability vector 10,000 times, and each time we correlated the shuffled vector to the second ROI’s unshuffled vector, and thus created a null distribution of correlation values. Next, we computed the difference between the ground-truth and the mean of the null distribution and divided it by the variance of the distribution, thus calculating a measure of effect size for each pair of ROIs. Note this is independent of the level of classification performance. To compare this effect size between deep and shallow conditions, paired t-tests with FDR correction were run across subjects. We found that for all pairs of ROIs, this effect size was statistically significantly greater in the deep compared to the shallow condition, except for the pair involving the precuneus and superior frontal gyrus ($p = 0.053$). This shows that the higher level of decoding accuracy in the deep processing condition can not account for the informational connectivity results.”

2. In response to my question about why left hemisphere ROIs were not included in the study, the authors stated: “The meta-analysis by Binder et al. (2009) identified 7 cortical regions involved in the processing of words. Binder et al. 2009 further reports evidence from the meta-analysis that the role of these regions in conceptual processing is left-lateralized.” The review by Binder et al. (2009) identified areas significantly associated with semantic language processing in both hemispheres, although a larger number of voxels reached significance in the left hemisphere (the red areas in Figure 9 of that paper). That is the only sense in which Binder et al. (2009) claim the activations are left-lateralized. Several studies published since then (some of which cited in the present manuscript) have confirmed that the right temporoparietal cortex, right precuneus/posterior cingulate, and portions of the right prefrontal cortex are involved in semantic word processing.

I do not think that excluding those right hemisphere areas from the present study would necessarily be a fatal flaw, but the authors should provide a stronger rationale for doing so. As language researchers, we must exercise care to avoid contributing to the widespread misconception that the right hemisphere does not contribute to language semantics.

We agree with the reviewer. We simply focused on the left hemisphere ROIs based on the preliminary evidence in order to constrain the search space of our fMRI analyses. We agree that it is important not underestimate the contribution of the right hemisphere to semantic processing. Accordingly we have added a note to this effect in the introduction when we refer to Binder meta-analysis. *“It is important to note that Binder et al. (2009) also identified brain areas associated with semantic language processing in the right hemisphere, although semantic-related activity was higher in the left hemisphere. Because of this, we elected to focus on key semantic regions of interest (ROIs) of the left hemisphere in order to constraint the search space of our fMRI analyses.”*